# Acquired cancer cell resistance to T cell bispecific antibodies and CAR T targeting HER2 through JAK2 down-modulation

Enrique J. Arenas [1,2,8], Alex Martínez-Sabadell [1,8], Irene Rius Ruiz [1,2], Macarena Román Alonso [1], Marta Escorihuela[1], Antonio Luque[1], Carlos Alberto Fajardo[3], Alena Gros[3], Christian Klein [4] & Joaquín Arribas [1,2,5,6,7 ✉]

Immunotherapy has raised high expectations in the treatment of virtually every cancer. Many current efforts are focused on ensuring the efficient delivery of active cytotoxic cells to tumors. It is assumed that, once these active cytotoxic cells are correctly engaged to cancer cells, they will unfailingly eliminate the latter, provided that inhibitory factors are in check. T cell bispecific antibodies (TCBs) and chimeric antigen receptors (CARs) offer an opportunity to test this assumption. Using TCB and CARs directed against HER2, here we show that disruption of interferon-gamma signaling confers resistance to killing by active T lymphocytes. The kinase JAK2, which transduces the signal initiated by interferon-gamma, is a component repeatedly disrupted in several independently generated resistant models. Our results unveil a seemingly widespread strategy used by cancer cells to resist clearance by redirected lymphocytes. In addition, they open the possibility that long-term inhibition of interferon-gamma signaling may impair the elimination phase of immunoediting and, thus, promote tumor progression.

[1] Preclinical Research Program, Vall d'Hebron Institute of Oncology (VHIO), Vall d'Hebron Barcelona Hospital Campus, Barcelona 08035, Spain. [2] Centro de Investigación Biomédica en Red de Cáncer (CIBERONC), Madrid 28029, Spain. [3] Tumor Immunology & Immunotherapy Group, VHIO, Vall d'Hebron Barcelona Hospital Campus, Barcelona 08035, Spain. [4] Roche Innovation Center Zurich, Roche Pharmaceutical Research and Early Development, Schlieren 8952, Switzerland. [5] Department of Biochemistry and Molecular Biology, Universitat Autónoma de Barcelona (UAB), Bellaterra 08193, Spain. [6] Cancer Research Program, IMIM (Hospital del Mar Medical Research Institute), Barcelona 08003, Spain. [7] Institució Catalana de Recerca i Estudis Avançats (ICREA), Barcelona 08010, Spain. [8] These authors contributed equally: Enrique J. Arenas, Alex Martínez-Sabadell. ✉email: jarribas@vhio.net

T cell-engaging therapies, such as T cell bispecific antibodies (TCBs)—which are functionally similar but structurally different to bispecific T cell engagers, BiTEs—or chimeric antigen receptors (CARs), are raising extraordinary expectations as future treatments for virtually all cancers (for recent reviews on the subject, see refs. [1–11]). Encouraging these expectations, TCBs and CARs have been recently approved to treat some hematologic malignancies[12–14]. In contrast, TCBs and CARs against solid tumors tested to date have failed to show clinical efficacy. This failure prompted intense research and the subsequent identification of mechanisms of primary and acquired resistance, including: (i) defective tumor infiltration of redirected lymphocytes, (ii) immunosuppressive tumor environments, (iii) downmodulation of the antigen against which TCB or CAR are directed and, (iv) upregulation of immune checkpoints (see, the aforementioned reviews). Different strategies are being implemented to overcome these mechanisms of resistance (reviewed in ref. [15]).

All these mechanisms of resistance impinge on the ability of T cells to reach cancer cells and/or on the inhibition of T cells. However, little is known about putative intrinsic mechanisms of resistance of cancer cells. That is, mechanisms deployed by tumor cells to resist killing by fully active and correctly engaged T cells. TCBs and CARs targeting the cell surface receptor HER2 are a useful tool to identify such mechanisms.

HER2 is a tyrosine kinase and, when overexpressed, a driver for breast and gastric cancers[16]. The downregulation of overexpressed HER2 may lead to tumor regression[17]. Thus, in principle, HER2-driven cancer cells are not likely to downmodulate the expression of HER2 to escape TCBs or CARs, making them a suitable experimental system to unveil mechanisms of intrinsic resistance to these therapies.

Using HER2-driven cell lines and patient derived xenografts (PDXs), and a TCB and CAR targeting HER2, here we describe experiments that unveil a mechanism of resistance to redirected T cells. This mechanism should be taken into consideration when designing strategies to increase the efficacy of cancer immunotherapies.

## Results

### Model of resistance to cell killing by redirected lymphocytes.
The structure of the HER2-TCB used in this study has been previously described[18] (see also Supplementary Fig. 1a). Addition of picomolar concentrations of this HER2-TCB to cocultures of peripheral blood mononuclear cells (PBMCs) and the HER2-overexpressing BT474 cells led to the killing of the latter (Supplementary Fig. 1b). It should be noted that the IC50 of HER2-TCB for a given ratio PBMC:BT474 varied depending on the donor of PBMCs. Similar differences have been previously observed[18], and are likely due to alloreactions of different intensities, which parallel different degrees of HLA mismatch, as well as T cell fitness that may depend on the donor. For the 1:1 ratio chosen for subsequent experiments, the IC50 of HER2-TCB varied from ~40 to ~220 pM.

To generate a model of intrinsic resistance to TCB-mediated cell killing, we treated PBMC:BT474 cocultures with HER2-TCB, allowed the cells to recover, and repeated the treatment with a fresh batch of PBMCs and HER2-TCB (Fig. 1a). At every round, we controlled that the alloreaction of PBMCs on target cells was low enough to allow the specific activity of the HER2-TCB. After 6 months, the resulting cells, BT-R, showed an IC50 for the HER2-TCB approximately tenfold higher than that of parental cells (156 pM vs. >1 nM) (Fig. 1b). Resistance was also observed in three dimensional cultures (Supplementary Fig. 1c) and in vivo (Fig. 1c, d). BT-R cells were also resistant to a trastuzumab-based second generation HER2-CAR (described in Supplementary Fig. 1a), in vitro and in vivo (Fig. 1e–h).

As expected, resistance was not due to the downregulation of HER2, as measured by Western blot or flow cytometry (Fig. 1i and Supplementary Fig. 1d). Further, the binding of HER2-TCB to parental and resistant cells was indistinguishable (Fig. 1j).

The sensitivity of resistant cells to different chemotherapeutic agents or to T-DM1, an antibody drug conjugate against HER2, was similar to that of parental BT474 cells (Supplementary Fig. 1e). Thus, BT-R cells are specifically resistant to killing by redirected lymphocytes.

We reasoned that resistance could arise by inhibition of T cells or, alternatively, by intrinsic resistance of target cells. Regarding the first possibility, analysis of a panel of immune checkpoint inhibitors and co-stimulators[19,20], showed no differences between parental and resistant cells, except the downmodulation of B7-H4 (Fig. 1k); however, since this is an inhibitory molecule[21], its downmodulation is not likely a cause of resistance. An array of 80 cytokines, including immunosuppressive cytokines such as TGFβ or IL10, revealed no major differences between cocultures containing parental or resistant cells. The only factor differentially secreted in cocultures with resistant cells was TIMP2 (Fig. 1l and Supplementary Fig. 2a). Quantification of TIMP2 levels confirmed that it is upregulated in BT-R cells (Supplementary Fig. 2b). To functionally characterize this observation, we downregulated the expression of TIMP2 by means of two specific short hairpin RNAs (shRNAs). The resulting cells, which expressed similar levels of TIMP2 to those of parental BT474 cells (Supplementary Fig. 2c), showed resistance to HER2-TCB similar to BT-R cells (Supplementary Fig. 2d). Thus, we concluded that the overexpression of TIMP2 by BT-R cells is unrelated to resistance.

Treatment of cocultures of PBMCs and parental BT474 cells with HER2-TCB augmented the secretion of several cytokines characteristic of lymphocyte activation, such as TNF-α, Interferon-gamma (IFN-γ), IL13, or IL5. Supporting that lymphocyte activation is unaffected by resistant cells, the secretion of theses cytokines was similar in assays including BT474 or BT-R cells (Supplementary Fig. 2e). Analysis of PBMC proliferation, or of different makers of activity, namely, CD69 and granzyme B, confirmed that activation of lymphocytes was unaffected in assays with resistant cells (Supplementary Fig. 2f, g). We concluded that lymphocytes are equally activated by resistant cells. Thus, we focused on possible mechanisms of intrinsic resistance of target cells.

### Transcriptomic analysis.
Analysis by RNA-seq showed that 97 genes were acutely upregulated or downregulated in BT-R cells compared to parental BT474 cells (≥4-fold; $p < 10^{-5}$) (Fig. 2a). Gene set enrichment analysis (GSEA) identified different biological processes that differed between parental and resistant cells (Fig. 2b). We focused in the IFN-γ response (Fig. 2c), because it has been shown that it affects the antitumor response at multiple levels[22]. Analysis of the protein encoded by *IRF1*, a gene specifically regulated by IFN-γ signaling[23], confirmed that this pathway is impaired in resistant cells (Fig. 2d).

### IFN-γ signaling is required for efficient killing by redirected lymphocytes.
To determine whether the downmodulation of IFN-γ signaling was related to resistance, first we used a blocking antibody. The anti-IFN-γ efficiently impaired killing of parental BT474 cells mediated by the HER2-TCB, in 2D or 3D cultures (Fig. 3a). Similarly, blocking IFN-γ prevented killing of target cells by HER2-CAR T cells (Fig. 3b).

IFN-γ acts on cytotoxic lymphocytes as well as on tumor cells. Maximal cytotoxic activity of T cells requires IFN-γ. On the other hand, IFN-γ impairs the proliferation of tumor cells by inhibiting their progression through the cell cycle and promoting their

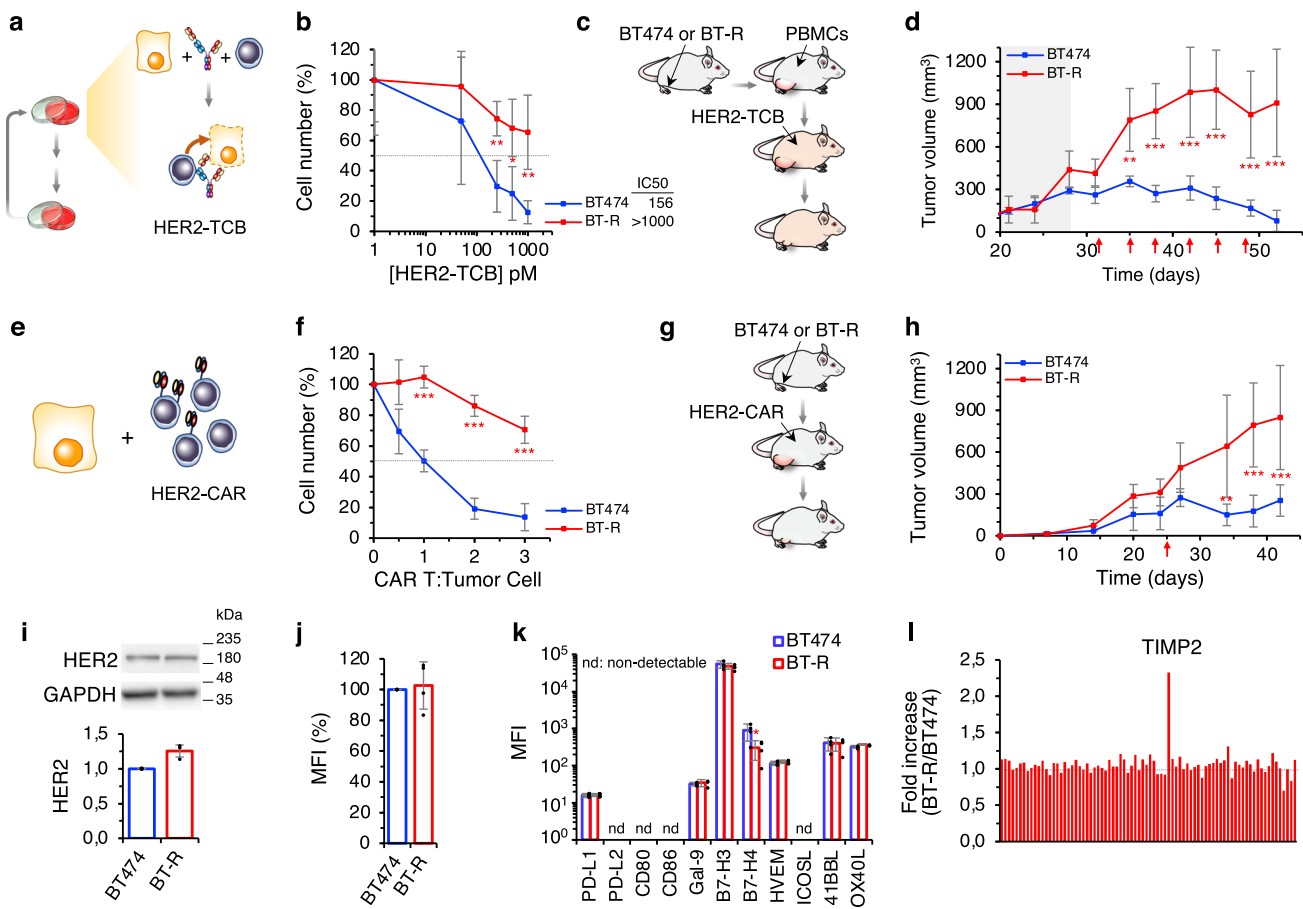

**Fig. 1 Generation and characterization of resistant cells. a** Schematic showing the assay of HER2-TCB in cocultures. **b** Cocultures of PBMCs with parental BT474 or resistant BT-R cells were treated with different concentrations of HER2-TCB (PBMC:target cell ratio 1:1) for 72 h. Then, viable cells were quantified by flow cytometry using EpCAM as a marker. **c** Schematic showing in vivo treatment with HER2-TCB. Totally, $10^7$ BT474 or BT-R cells were injected orthotopically into NSG mice. When tumors reached ~200 mm³ (dark background), $10^7$ PBMCs were injected i.p. Then animals were treated i.v. with 0.125 mg/kg HER2-TCB as indicated (red arrows). **d** Tumor volumes are represented as averages ± SD (standard deviation) ($n = 4$ per arm). **e** Schematic showing the HER2-CAR assay. **f** Parental BT474 cells or resistant BT-R cells were cocultured with different ratios of CAR T cells for 48 h. Cell numbers were calculated and expressed as in (**b**). **g** Schematic showing in vivo treatment with HER2-CAR T cells. Mice were injected with BT474 or BT-R cells as in (**c**). When tumors reached ~200 mm³, $3 \times 10^6$ HER2-CAR T-positive cells were injected i.p. **h** Tumor volumes are represented as averages ± SD (BT474, $n = 6$; BT-R, $n = 8$). **i** Levels of HER2, normalized to BT474 cells, were determined by Western Blot. **j** Cells were stained with HER2-TCB and analyzed by flow cytometry. Results were normalized to BT474 cells. **k** Cells were stained with antibodies against the indicated factors. Results are presented as the MFI of staining in BT474 and BT-R cells. **l**, Levels of 80 cytokines and growth factors were measured using a commercial array in the media conditioned by cocultures of BT474 cells and PBMCs or BT-R cells and PBMCs treated with HER2-TCB. Results are presented as the fold change of BT-R relative to BT474 cells. **b** **p = 0.005, *p = 0.02, **p = 0.006. **f** ***p < 0.001. **k** *p = 0.04, two-tailed t test. **d, h** **p < 0.01, ***p < 0.001, two-way analysis of variance (ANOVA) and Bonferroni correction. Data are presented as mean ± SD of four (**b, f, j, k**), or three (**i**) independent experiments. Source data are provided as a Source Data file.

apoptotic death (reviewed in ref. [22]). Thus, the blocking antibody could prevent the killing of target cells by impairing the activation of cytotoxic lymphocytes, blocking the action of IFN-γ on target cells, or a combination of both effects.

To directly test the effect on target cells, we treated BT474 cells with IFN-γ. As shown in Fig. 3c, IFN-γ induced the death of BT474 cells by inducing apoptosis, as shown in the increase of annexin V⁺ cells (Fig. 3d). In contrast, resistant cells were unaffected by the same treatment. This result shows that IFN-γ is likely part of the mechanism used by cytotoxic cells to kill target cells, and indicates that BT-R cells became resistant by impairing IFN-γ signaling.

To confirm this hypothesis, we impaired IFN-γ signaling in target cells, by knocking-out the IFNGR1 gene through CRISPR–Cas9 technology, or by downmodulating the expression of IFNGR1 with shRNAs from parental BT474 cells (Fig. 3e). As expected, both methods impaired IFN-γ signaling (Supplementary

Fig. 3a, b), and prevented the killing induced by IFN-γ (Fig. 3f). Irrespectively of the method, diminishing the expression of IFNGR1 resulted in resistance to HER2-TCB in coculture assays (Fig. 3g). Consistent results were obtained when we examined the sensitivity to HER2-CAR T cells (Fig. 3h). To validate the results obtained in vitro, we showed that the knock-out of IFNGR1 caused resistance to the HER2-TCB also in vivo (Fig. 3i). The knockdowns of JAK1 or STAT1 from parental BT474 cells also resulted in resistance to the HER2-TCB (Supplementary Fig. 3c, d). Thus, disrupting IFN-γ signaling by different means induces resistance to redirected lymphocytes.

The effects of the IFN-γ-blocking antibodies or of the knock-down of IFNGR1 were not a particularity of BT474 cells, similar effects were observed when assaying different HER2-positive cultures from breast cancer PDXs and an additional cell line (Supplementary Fig. 4a–i). Furthermore, impairment of IFN-γ

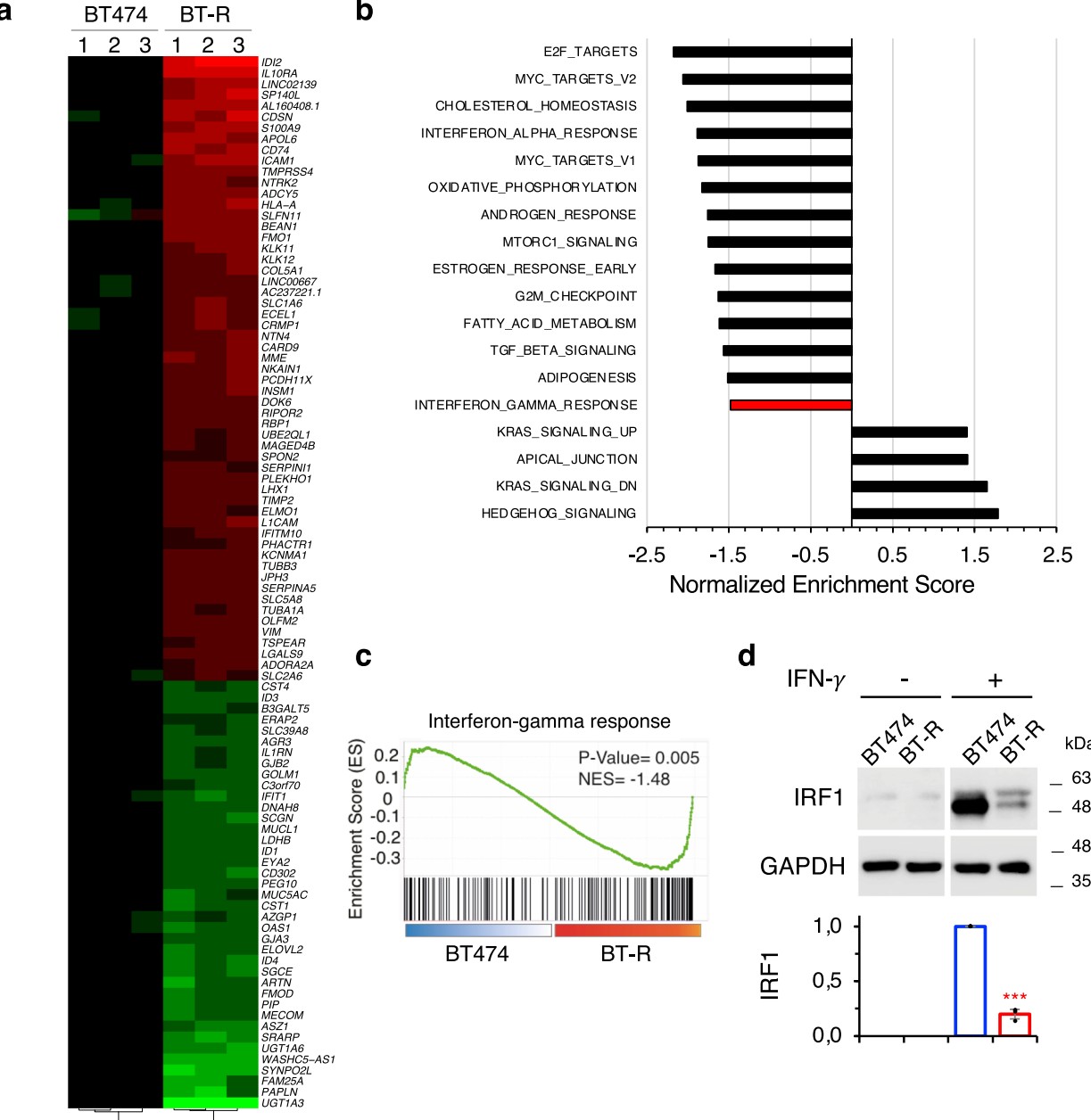

**Fig. 2 Transcriptomic analysis. a** Heatmap showing the 97 most differentially expressed genes in BT-R cells compared to parental BT474 cells (≥4-fold; $p < 10^{-5}$). $p$ Value was obtained from the DESeq2 analysis of the RNA-seq. Genes with FDR < 5% and |shrunken fold change| > 1.5 were considered significant. **b** Pathways showing positive and negative enrichment in BT-R compared with BT474 cells as determined by GSEA. Only statistical gene sets are shown (NOM $p$ value < 0.05). **c** IFN-γ response GSEA signature in resistant cells, compared to parental BT474. $p$ Value corresponds to the NOM $p$ value obtained by GSEA in the HALLMARK database. **d** Levels of IRF1 upon treatment with IFN-γ in BT474 and BT-R cells were determined by Western blot analysis. Results of three independent quantifications, normalized to treated BT474 cells, are presented as mean ± SD. ***$p < 0.001$, two-tailed $t$ test. Source data are provided as a Source Data file.

signaling by knocking down IFNGR1, led to resistance to HER2-TCB in cell lines from HER2-positive ovarian and lung cancers (Supplementary Fig. 4j, k).

**Components of the IFN-γ signaling pathway in resistant cells.** The intracellular signaling pathway activated by IFN-γ is relatively simple. Dimeric IFN-γ triggers the formation of a complex of four receptor molecules (two IFNGR1 and two IFNGR2), four JAK kinases (two JAK1s and two JAK2s) and two molecules of the STAT1 transcription factor. The subsequent phosphorylation of the STAT1s by the JAKs leads to the dimerization and transport to the nucleus of the former, where dimeric pSTAT1 regulates the expression of different genes, including IRF1[22] (Fig. 4a). Analysis of these components in our coculture assays revealed little or no difference in the secretion of IFN-γ or in the expression of IFNGR1 and 2, JAK1 or STAT1 by resistant cells (Fig. 4b–e). In contrast, we observed a marked reduction of JAK2 expression (Fig. 4f), indicating that downmodulation of JAK2 could result in the impairment of IFN-γ signaling. Consistently with this hypothesis, pSTAT1 levels were significantly decreased in resistant cells when treated with IFN-γ (Fig. 4g).

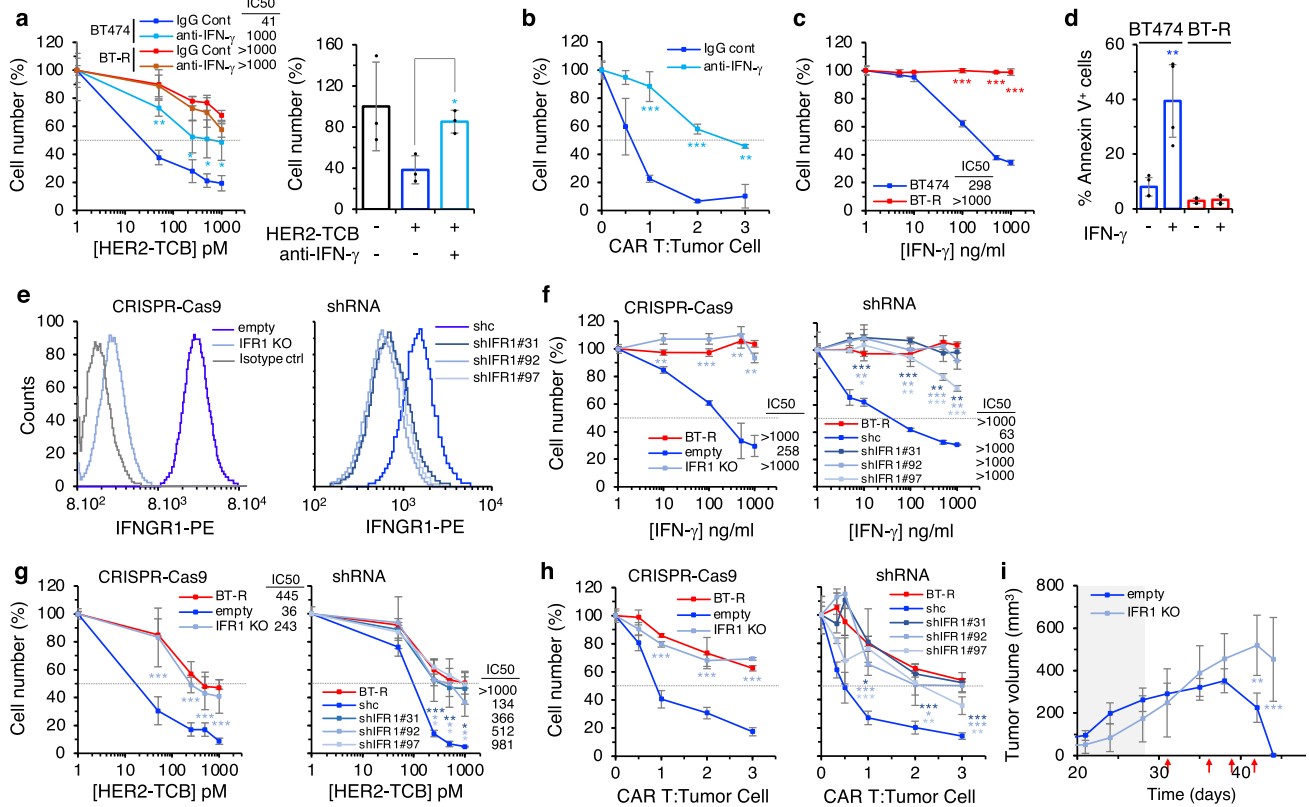

**Fig. 3 IFN-γ response is required for efficient killing by redirected lymphocytes. a** Left, cocultures of PBMCs with BT474 or BT-R cells were treated with different concentrations of HER2-TCB in presence of an IgG control or an IFN-γ blocking antibody for 72 h. Then, viable cells were quantified by flow cytometry using EpCAM as a marker. Right, BT474 cells were grown in 3D and treated with HER2-TCB in presence of an IgG control (−) or an IFN-γ blocking antibody (+) for 72 h. Viable BT474 cells were quantified by flow cytometry using EpCAM as a marker. Results were normalized to untreated cells. **b** Parental BT474 cells were cocultured with different ratios of CAR T cells for 48 h in the presence of an IgG control or an IFN-γ blocking antibody. Then, cell numbers were calculated and expressed as in (**a**). **c** Parental BT474 or resistant BT-R cells were treated with different concentrations of IFN-γ for 5 days. Cell numbers were estimated with the crystal violet staining assay. **d** Parental BT474 or resistant BT-R cells were treated with 1 μg/ml of IFN-γ and the percentages of apoptotic cells were determined by means of Annexin V+ cells measured by flow cytometry. **e** Cells were stained with anti-IFNGR1 or isotype antibody and analyzed by flow cytometry. IFR1 stands for IFNGR1. **f** Sensitivity of the indicated cells to IFN-γ was analyzed as in (**c**). **g** The indicated cells were treated with different concentrations of HER2-TCB and analyzed as in (**a**). **h** The indicated cell lines were cocultured with CAR Ts and analyzed as in (**b**). **i** Totally, $6.5 \times 10^7$ BT474 cells or the same cells knock-out for IFNGR1 were injected orthotopically into NSG mice. Mice were treated as described in Fig. 1c. Tumor volumes are represented as averages ± SD (empty, n = 8; IFR1 KO, n = 4). **a** Left, **p = 0.001, *p = 0.05; right, *p = 0.01. **b** ***p < 0.001, **p = 0.002. **c** ***p < 0.001. **d** **p = 0.007. **f** Left, **p = 0.002, ***p < 0.001, **p = 0.003, **p = 0.0013; right, ***p < 0.001, **p = 0.002 (shIFNGR1 #31); **p = 0.009, **p = 0.006, ***p < 0.001, **p = 0.003 (shIFNGR1 #92); *p = 0.02, **p = 0.004, ***p < 0.001 (shIFNGR1 #97). **g** Left, ***p < 0.001; right, ***p < 0.001, **p = 0.004, *p = 0.02 (shIFNGR1 #31); *p = 0.046, *p = 0.047, *p = 0.03 (shIFNGR1 #92); *p = 0.05, *p = 0.02, *p = 0.02 (shIFNGR1 #97). **h** Left, ***p < 0.001; right *p = 0.02, ***p < 0.001 (shIFNGR1 #31); ***p < 0.001, *p = 0.02 (shIFNGR1 #92); ***p < 0.001, **p = 0.005, **p = 0.005 (shIFNGR1 #97). Two-tailed t test. **i** **p < 0.01, ***p < 0.001, two-way ANOVA and Bonferroni correction. Data are presented as mean ± SD of three (**a-c**, **f**, **g** right, **h**), four (**d**), or six (**g** left) independent experiments. Source data are provided as a Source Data file.

Resistant cells cultured for up to three months in the absence of any selective pressure showed resistance similar to that of BT-R cells (Supplementary Fig. 5a) and similar levels of JAK2 (Supplementary Fig. 5b). Therefore, resistance is stable and does not require selective pressure. We hypothesized that this stable downmodulation of JAK2 and, thus, resistance could be maintained epigenetically. To test this hypothesis, we treated BT-R cells with different well-characterized drugs that interfere with epigenetic modifications, including the DNA demethylating agent 5-AZA, the histone demethylating agent DZNEP, and the pan-HDAC inhibitor Trichostatin A (TSA). While demethylating agents did not affect *JAK2* transcription, the HDAC inhibitor upregulated *JAK2* expression in BT-R cells (Fig. 4h), indicating that de-acetylation of histone H3 may contribute to the silencing of *JAK2*. Direct analysis of H3K27Me3 and H3K27Ac marks showed reduced levels of the latter in the *JAK2* promoter of BT-R cells (Fig. 4i), confirming that histone acetylation may regulate

JAK2 expression. Among the different acetylation marks, H3K27Ac is considered one of the best markers of active promoters and enhancers and it is tightly associated to gene expression[24]. Thus, we concluded that JAK2 is likely down-regulated epigenetically in BT-R cells.

**The downmodulation of JAK2 causes resistance to redirected lymphocytes**. To establish the functional relevance of the downmodulation observed, we transduced resistant cells with a vector encoding JAK2 (Fig. 5a). As expected, overexpression of JAK2 restored IFN-γ signaling (Supplementary Fig. 5c). Of note, overexpression of JAK2 resulted in re-sensitization to killing by T cells redirected via TCB or CAR and in the re-sensitization to the death induced by IFN-γ (Fig. 5b–d). Confirming the relevance of these in vitro results, overexpression of JAK2 restored sensitivity to the TCB in vivo (Fig. 5e).

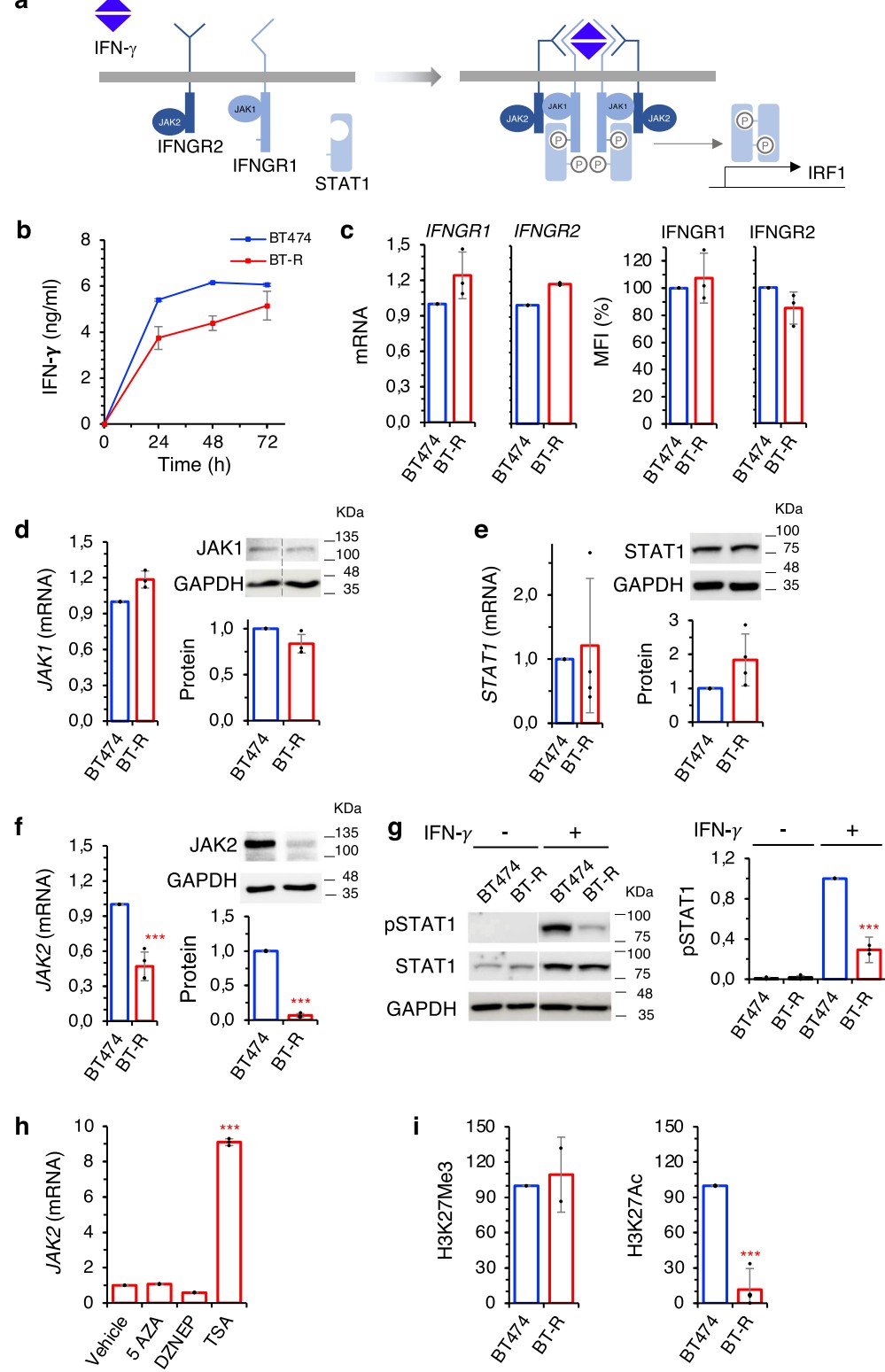

**Fig. 4 Components of the IFN-γ signaling pathway in resistant cells. a** Schematic showing the IFN-γ signaling pathway. **b** The levels of IFN-γ in the media conditioned by cocultures with BT474 or resistant BT-R in the presence of HER2-TCB was determined by ELISA. **c** The levels of Interferon-gamma receptors 1 and 2, normalized to BT474, were determined by quantitative real-time PCR (left) or flow cytometry with specific antibodies (right). **d–f** Levels of JAK1, STAT1, and JAK2 (mRNA and protein), normalized to BT474, as determined by quantitative real-time PCR (left) or Western blot (right). **g** Levels of phosphoSTAT1 in parental and resistant cells were determined by Western Blot. Results were normalized to treated BT474 cells. **h** BT-R cells were treated with indicated compounds for 48 h. Then, the levels of the mRNA encoding *JAK2* were determined by RT-PCR and normalized to the levels in cells treated with vehicle. **i** Levels of H3K27Me3 and H3K27Ac histone marks in the promoter of *JAK2* in BT474 and BT-R cells as measured by ChIP followed by quantitative real-time PCR. Results were normalized to the levels of an IgG control antibody in each sample, and then normalized to the levels in BT474 cells. ***$p < 0.001$, two-tailed *t* test. Data are presented as mean ± SD of two (**b**, **i** left), three (**c**, **d**, **h**, **i** right) or four (**e–g**) independent experiments. Source data are provided as a Source Data file.

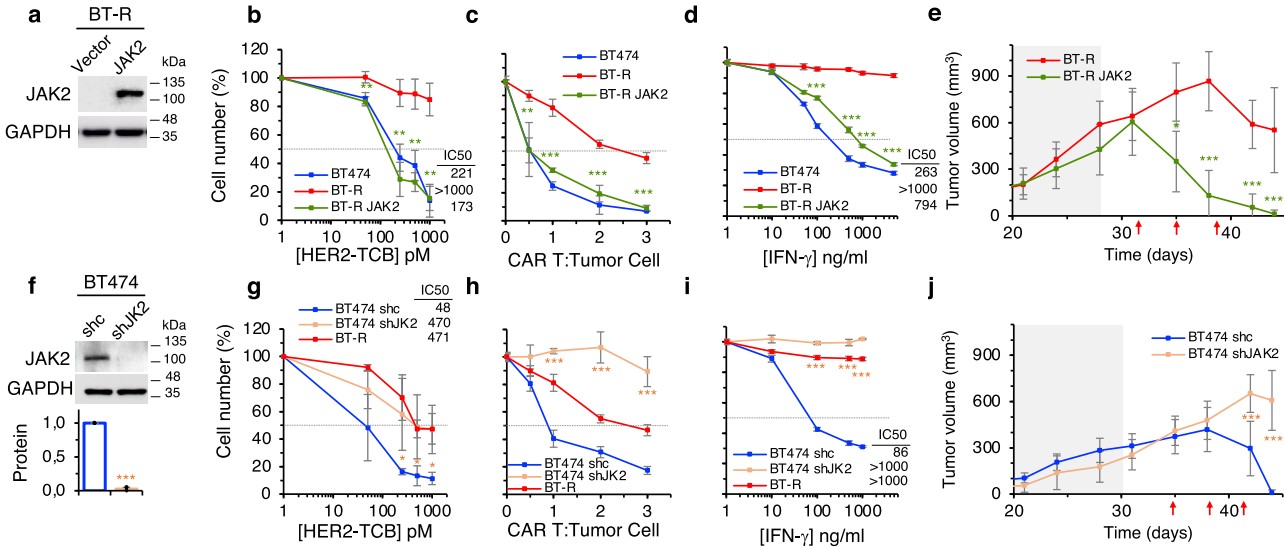

**Fig. 5 The downmodulation of JAK2 causes resistance to redirected lymphocytes. a** Levels of JAK2 in BT-R cells or the same cells stably transfected with a vector encoding JAK2 were determined by Western blot. Blot shown is representative of four independent experiments. **b** Cocultures of PBMCs with the indicated cells were treated with different concentrations of HER2-TCB for 72 h. Then, viable target cells were quantified by flow cytometry using EpCAM as a marker. **c** The indicated cells were treated with different ratios of CAR T cells. Cell numbers were calculated and expressed as in (**b**). **d** The indicated cells were treated with different concentrations of IFN-γ for 5 days. Cell numbers were estimated with the crystal violet staining assay. **e** Totally, 6.5 × 10⁶ BT-R cells or the same cells expressing JAK2 were injected orthotopically into NSG mice. When tumors reached ~200 mm³ (dark background), 10⁷ PBMCs were injected i.p. Then animals were treated i.v. with 0.125 mg/kg HER2-TCB as indicated (red arrows). Tumor volumes are represented as averages ± SD (n = 7 per arm). **f** Levels of JAK2 in BT474 cells stably expressing a non-targeting shRNA (shc) or an shRNA targeting JAK2 (shJK2) were determined by Western blot. Results were normalized to BT474 shc. **g–i** The indicated cells were analyzed as in (**b–d**), respectively. **j** Totally, 6.5 × 10⁶ BT474 cells stably expressing a non-targeting shRNA (shc) or an shRNA targeting JAK2 (shJAK2) were injected into NSG mice and treated as in (**e**). Tumor volumes are represented as averages ± SD (shc, n = 10; shJAK2, n = 5). **b** **p = 0.005, **p = 0.003, **p = 0.002, **p = 0.002. **c** **p = 0.002, ***p < 0.001. **d** ***p < 0.001. **g** *p = 0.05, *p = 0.05, *p = 0.02. **h**, **i** ***p < 0.001, two-tailed t test. **e**, **j** *p < 0.05, ***p < 0.001, two-way ANOVA and Bonferroni correction. Data are presented as mean ± SD of three (**b–d**, **h**, **i**) or four (**f**, **g**) independent experiments. Source data are provided as a Source Data file.

Conversely, knock-down of JAK2 from parental cells impaired IFN-γ signaling (Fig. 5f and Supplementary Fig. 5d), induced resistance to redirected lymphocytes and to cell death induced by IFN-γ in vitro (Fig. 5g–i), and resistance to the HER2-TCB in vivo (Fig. 5j).

Conceivably, impairment of any of the components that transduce its signal results in defective response to IFN-γ. To determine if different components can be downmodulated in different resistant models, we generated an independent resistant BT474 model following an in vivo approach. We recovered cells from the residual tumor that remained after TCB-treatment, regrafted them into mice and repeated the process (Supplementary Fig. 6a). After the second round of selection, we obtained cells resistant to HER2-TCB, which were named BT-vR (Supplementary Fig. 6b). The levels of cell surface HER2 in resistant cells were unaltered (Supplementary Fig. 6c), and we observed a downmodulation of JAK2 similar to that in the in vitro selected BT-R cells (Supplementary Fig. 6d). Lymphocyte infiltration, stained with anti-CD3 antibodies, was similar in tumors generated by parental and resistant cells (Supplementary Fig. 6e), showing that deficient infiltration of lymphocytes was not the cause of resistance. Finally, in contrast with parental cells, IRF1 was undetectable in resistant cells (Supplementary Fig. 6e), confirming defective IFN-γ signaling in resistant cells.

To select resistant cells from another source and in different conditions, we implanted a HER2-positive PDXs in mice humanized with CD34⁺ hematopoietic stem cells as previously described[18]. Once the tumors reached ~200 mm³, mice were treated with the HER2-TCB, the tumors regressed, were allowed to regrow and the treatment was repeated (Supplementary Fig. 6f). After two rounds of treatment, the tumors were no longer sensitive to the HER2-TCB (Supplementary Fig. 6g). Again, the resistant PDX showed unaltered levels of HER2 but strongly decreased levels of JAK2 (Supplementary Fig. 6h, i). Thus, resistance to HER2-TCB in different contexts results in the same defect: downmodulation of JAK2 and impaired IFN-γ signaling in tumor cells.

**Models of resistance to IFN-γ.** Treatment of BT474 cells with increasing concentrations of IFN-γ starting at the IC50 during 4 months resulted in resistant cells (designed BT-RG), with an IC50 comparable to that of cells resistant to the HER2-TCB (BT-R) (>1000 ng/ml) (Fig. 6a). As expected, IFN-γ signaling was downmodulated in these resistant cells (Supplementary Fig. 5e). In vitro assays showed that IFN-γ resistant cells were also resistant to the HER2-TCB and to HER2-CAR T cells (Fig. 6b, c). Further, IFN-γ resistant cells were also resistant to the HER2-TCB in vivo (Fig. 6d). Thus, cells selected because of their resistance to IFN-γ showed similar characteristics to those selected for resistance to TCBs, further supporting the relevance of the IFN-γ pathway in resistance to redirected lymphocytes.

Importantly, as was the case with HER2-TCB resistant cells, the levels of JAK2 were significantly reduced in BT-RG cells (Fig. 6e). Next, we performed gain-of-function experiments; that is, analysis of IFN-γ signaling (Supplementary Fig. 5f) and the sensitivity to IFN-γ, HER2-TCB and HER2-CAR of BT-RG cells transfected with JAK2 (Fig. 6f–h). The results confirmed the causal role of JAK2 downmodulation on resistance to killing by redirected lymphocytes, independently of the method to generate cells resistant to IFN-γ signaling.

Similarly to BT-R cells, treatment of BT-RG cells with drugs that interfere with epigenetic modifications showed an effect of the pan-HDAC inhibitor Trichostatin A (TSA) on the expression of *JAK2* (Fig. 6i). In addition, BT-RG showed reduced levels of H3K27Ac in the promoter of *JAK2* (Fig. 6j). Thus, we concluded that JAK2 is also downregulated epigenetically in BT-RG cells.

Chronic treatment with IFN-γ is a convenient way to obtain models of resistance. Thus, to confirm our conclusions, we treated cultures from the two HER2-postive PDXs with IFN-γ to obtain resistant cells. In both cases, resistance to IFN-γ was acquired through downmodulation of JAK2. Consistent with the results obtained with BT474 cells, IFN-γ resistance induced resistance to killing by TCB; further, JAK2 overexpression rescued completely the phenotype (Supplementary Fig. 7). We concluded that breast cancer cells acquire resistance to IFN-γ and, thus, resistance to redirected lymphocytes, by downregulating JAK2 (Fig. 6k).

## Discussion

Redirection of lymphocytes, via TCBs or CARs, is already approved to treat some hematological malignancies. This success contrasts with the failures in the treatment of solid tumors. In order to overcome this lack of efficacy, many efforts have focused on counteracting the inhibitory effect that the tumor microenvironment exerts on lymphocytes. Only recently, intrinsic mechanisms or resistance, such as defective death receptor signaling causing resistance to CARs directed against CD19, have been described[25]. The simple approach used in this study allowed to unveil additional mechanisms deployed by cancer cells to resist killing by a fully active lymphocyte with unrestricted access to its target cell.

The impairment of the IFN-γ pathway does not have discernible effects on cell proliferation, as readily shown by the knockdown or knockout of IFNGR1, JAK1, STAT1, or JAK2, but severely affects the sensitivity of cells to the killing by redirected lymphocytes. Thus, alterations conducting to impairment of IFN-γ signaling in tumor cells are likely to arise in patients under the selective pressure imposed by TCBs or CARs.

Activated T cells engage target cells via a cytolytic synapse formed between the TCR and the MHC-antigen complex. These synapses cause the death of the target cells via perforin- and granzyme-induced apoptosis. In addition, FAS-L produced by activated lymphocytes causes death of cells expressing the FAS receptor[26,27]. It is generally assumed that this system suffices to kill target cells and that TCBs and CARs trigger the same mechanism. The results presented here show a previously unnoticed role of the IFN-γ pathway in the killing of HER2-positive cancer cells by lymphocytes, at least those redirected via TCBs or CARs targeting HER2.

Compared with TCR/MHC-antigen synapses, the contacts mediated by TCBs or CARs have some particularities (reviewed in ref. [4]). First, contacts formed via TCR or CARs function independently of MHC[28]. On the other hand, the affinities of TCRs for TCR/MHC-antigen complexes are in the range of 1–100 μM, whereas the affinities of TCBs and CARs are typically below 100–10 nM[29,30]. Finally, few complexes of a given antigen with its cognate MHC are expressed per target cells[31]; in contrast, targets of TCBs and CARs may number in the 1000–10,000 s, or in the 100,000 s in the case of HER2-amplified tumors[32]. In fact, differences between the synapses established by TCR/MHC-antigen and CARs have been directly shown[33]. Probably because of these differences, while the activation of T cells by TCR/MHC–antigen synapses requires additional second signals, TCBs and CARs suffice to activate lymphocytes. Future work should clarify if the dependence of IFN-γ signaling is a particularity of T cell redirected via TCBs or CARs against cells expressing high levels of an antigen, or if it also applies to T cells engaged through TCR/MHC-antigen synapses.

In this regard, loss-of-function mutations in JAK1 or JAK2 have been shown to arise in tumors that progressed to treatment with immune checkpoint inhibitors, showing that IFN-γ pathway may also be critical for the killing of target cells by the TCR/MHC–antigen complex. In addition, recently the downmodulation of genes upregulated by IFN-γ has been correlated with resistance to immune checkpoint blockade[33,34], further supporting the relevance of IFN-γ signaling on the sensitivity of target cells to killing by cytotoxic lymphocytes.

The results presented here, have practical implications, particularly when considering the widespread use of JAK2 inhibitors in the clinic[34]. Our results imply that the systemic use of these inhibitors, currently approved to treat myelofibrosis and hydroxyurea resistant or intolerant polycythemia vera to alleviate the exacerbated activation of the immune system, may impact on cancer immunoediting and, in some contexts, favor tumor progression.

## Methods

**Study design.** This study was designed to identify the mechanisms that trigger the resistance against HER2-TCB and -CAR. In order to generate acquired resistance models and to identify the mechanism described in this work, in vitro and in vivo functional assays were performed, using both tumor cell lines and PDXs. Experiments consisted of combining tumor cells with human lymphocytes and TCBs or CARs targeting HER2. For in vivo models, two sources of human immune cells were used: PBMCs and CD34+ hematopoietic stem cells. All human samples were obtained with informed consent and following institutional guidelines under protocols approved by the Institutional Review Boards (IRBs) at Vall d'Hebron Hospital. Animal work was performed according to protocols approved by the Ethical Committee for the Use of Experimental Animals at the Vall d'Hebron Institute of Oncology. For in vivo experiments, two to five mice were used per group, and they were randomized by tumor size. Mice that died before the end of the experiments for reasons unrelated to treatment or that did not have detectable percentages of human immune cells were excluded. Because of ethical reasons, we ended the experiments before the full development of graft-versus-host disease or when tumor volume surpassed 1500 mm3. Experiments were performed in a blinded fashion.

**Cell lines and primary cultures.** BT474 (#HTB-20), SKBR3 (#HTB-30), HEK293T (#CRL-11268), SKOV3 (#HTB-77), and H1781 (#CRL-5894) were obtained from ATCC (Manassas, VA, USA). GP2-293 cells (#631458) were obtained from Clontech. PDX433 (ER+/PR−/HER23+) was extracted by core needle biopsy (CNB) from a breast cancer patient's metastasis in the liver. PDX667 (HER2+) was extirpated by CNB from a breast cancer patient's metastasis in the skin. PDX118 (ER+/PR−/HER23+) comes from a skin metastasis collected by CNB. All these PDXs have been established at VHIO following institutional guidelines. The IRBs at Vall d'Hebron Hospital provided approval for this study in accordance with the Declaration of Helsinki. Written informed consent was obtained from all patients who provided tissue samples.

Cell lines were cultured under standard conditions in complete medium (DMEM F-12 medium (#21331, Gibco) supplemented with 10% fetal bovine serum (FBS) (#10270, Gibco), 1% L-glutamine (#X0550, Biowest), and 1% penicillin–streptomycin (#P4333, Sigma-Aldrich)). Cells were genetically modified to acquire resistance to certain antibiotics or to downmodulate or overexpress different genes. Cells were routinely tested for the absence of mycoplasma contamination using the MycoAlert™ Mycoplasma detection kit (#LT07-318, Lonza). Cell lines were not authenticated in-house.

**PBMC isolation.** PBMCs were isolated from fresh buffy coats obtained from healthy donors through the Blood and Issue Bank of Catalonia (BST). Blood was diluted 1:3 with 1× phosphate-buffered saline (PBS) and transferred to a 50 ml falcon tube with Ficoll-Paque PLUS (#70-1440-02, GE Healthcare) at a 1:3 ratio, following the manufacturer's instructions. After obtaining the buffy coat, red blood cells (RBC) were lysed with 1× RBC lysis buffer (#00-4333-57, Invitrogen) for 4 min. Obtained PBMCs were counted and frozen with Cryostor CS10 (#07959, Stemcell Technologies) at −80 °C for in vitro and in vivo experiments.

**Generation of resistant cells in vitro.** To generate the BT-R model, BT474 cells stably expressing hygromycin resistance were treated with a 3:1 ratio PBMC:Tumor and a concentration of 67.5 pM HER2-TCB in PBMC media (RPMI 1640 (#61870, Gibco), 10% heat-inactivated FBS, 1% L-glutamine, 1% HEPES (#H0887, Sigma-Aldrich), 1% MEM nonessential amino acids solution (#11140, Gibco), and 1% penicillin–streptomycin). After 72 h, media was removed and replaced with complete medium containing 100 μg/ml hygromycin (#10687010, Gibco) during 7 days

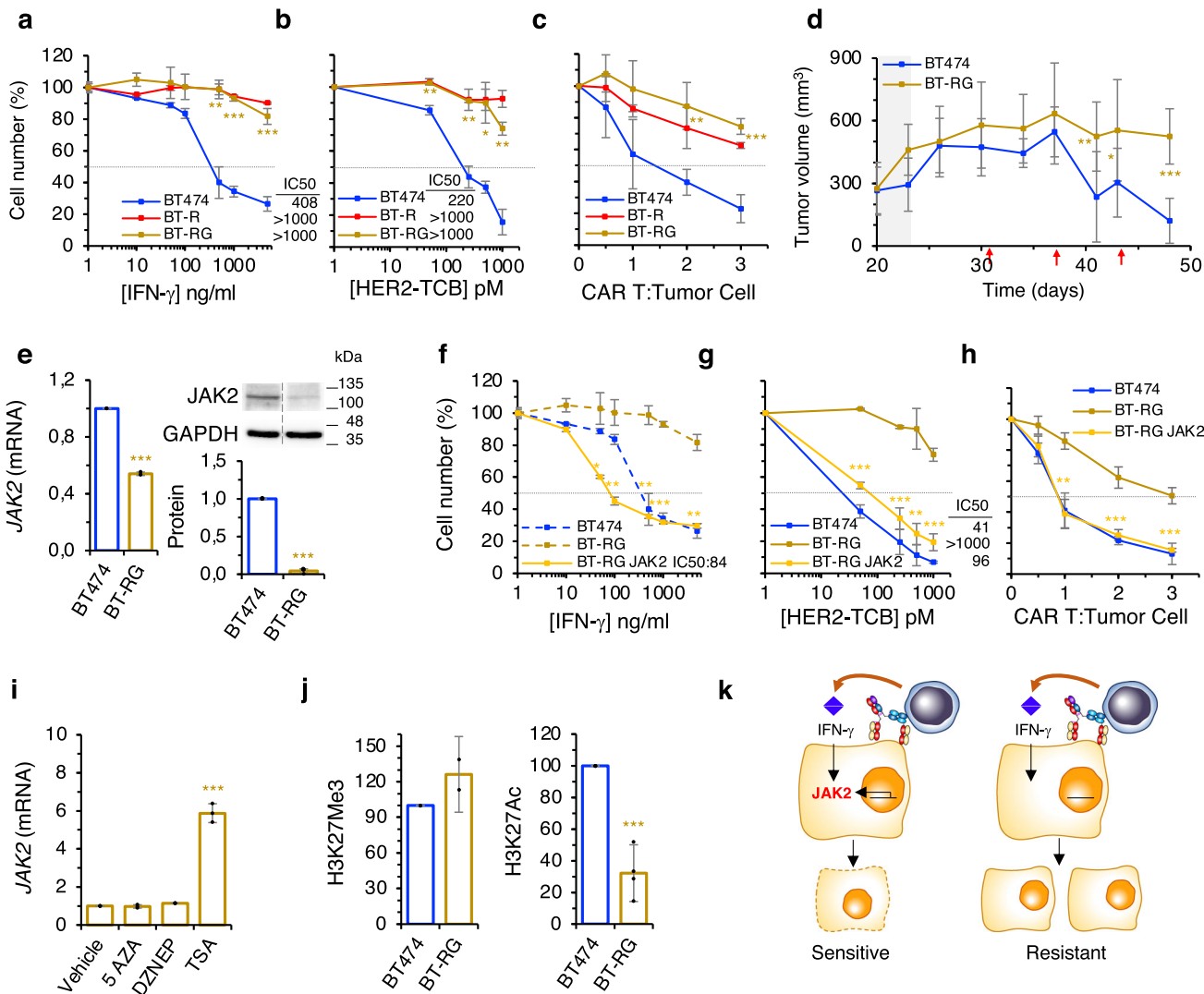

**Fig. 6 Model of resistance to IFN-γ. a** Parental BT474 cells or resistant BT-R or BT-RG cells were treated with different concentrations of IFN-γ for 5 days. Cell numbers were estimated with the crystal violet staining assay. **b** Cocultures of PBMCs with the indicated cells were treated with different concentrations of HER2-TCB for 72 h. Then, viable target cells were quantified by flow cytometry using EpCAM as a marker. **c** The same cells as in (**a**) were treated with different ratios of CAR T:Tumor cell. Cell numbers were obtained by means of EpCAM counts by flow cytometry. **d** Totally, $10^7$ BT474 or resistant BT-RG cells were injected orthotopically into NSG mice. When tumors reached ~300 mm$^3$ (dark background), $10^7$ PBMCs were injected i.p. Then animals were treated i.v. with 0.125 mg/kg HER2-TCB as indicated (red arrows). Tumor volumes are represented as averages ± SD (BT474, $n = 8$; BT-RG, $n = 6$). **e** The levels of JAK2 (mRNA and protein) as determined by quantitative real-time PCR (left) or Western blot (right). Results were normalized to BT474 cells. **f–h** The indicated cells were analyzed as in (**a–c**), respectively. **i** BT-RG cells were treated with indicated compounds for 48 h. Then, the levels of the mRNA encoding JAK2 were determined by RT-PCR and normalized to the levels in cells treated with vehicle. **j** Levels of H3K27Me3 and H3K27Ac histone marks in the promoter of JAK2 in BT474 and BT-RG cells as measured by ChIP followed by quantitative real-time PCR. Results were normalized to the levels of an IgG control antibody in each sample, and then normalized to the levels in BT474 cells. **k** Schematic drawing summarizing our findings. **a** **$p = 0.002$, ***$p < 0.001$. **b** **$p = 0.009$, **$p = 0.006$, *$p = 0.013$, **$p = 0.002$. **c** **$p = 0.008$, ***$p < 0.001$. **e** ***$p < 0.001$. **f** *$p = 0.02$, **$p = 0.004$, **$p = 0.002$, ***$p < 0.001$, **$p = 0.002$. **g** **$p = 0.005$, ***$p < 0.001$. **h** **$p = 0.0015$, ***$p < 0.001$. **i, j** ***$p < 0.001$. Two-tailed $t$ test. **d** *$p < 0.05$, **$p < 0.01$, ***$p < 0.001$, two-way ANOVA and Bonferroni correction. Data are presented as mean ± SD of three (**a–c, e–i**), two (**j** left) or four (**j** right) independent experiments. Source data are provided as a Source Data file.

to specifically kill remaining PBMCs. The process was repeated several times. Resistant population was obtained after 6 months.

Interferon-gamma resistant models were established by culturing cells in presence of increasing IFN-γ (#300-02, Peprotech) concentrations, starting at 100 ng/ml and reaching 1 μg/ml. In the three models (BT474-RG, PDX667-RG, and PDX433-RG) resistance was obtained after 4 months of treatment.

**T cell cytotoxicity assays.** All target cells were seeded in 96-well flat bottom plates (0.01 × 10$^6$ cells/well) (#353075, Corning Life Sciences). Effector PBMCs were added to each well at the indicated ratio in PBMC medium. Different concentrations of HER2-TCB were added to the wells. The plates were incubated for 72 h.

At the endpoint cells were harvested with trypsin–EDTA (#25300096, Gibco) and resuspended in fluorescence-activated cell sorting (FACS) buffer (PBS 1×,

2.5 mM EDTA, 1% bovine serum albumin (BSA) (#A9647, Sigma-Aldrich), 5% horse serum (#26050, Gibco)) in polypropylene V-bottom 96-well plates (#651201, Greiner Bio-One). Twenty minutes later, samples were centrifuged and cells were stained with the epithelial cell marker anti-human EpCAM (#324212, BioLegend) at 1:300 concentration in FACS buffer in ice for 30 min. After a wash with 1× PBS, samples were resuspended in the viability marker Zombie Aqua at 1:1000 (#423101, BioLegend) in 1x PBS and acquired on LSR Fortessa, using BD FACSDiva software (BD Biosciences). Number of alive cells was analyzed with FlowJo software (BD Life Sciences) by means of EpCAM counts. Gating strategy for tumor EpCAM$^+$ counts is shown in Supplementary Fig. 8.

When treated with CAR Ts, target cells were seeded as described above. After 24 h, effector CAR Ts targeting HER2 were added to each well at the indicated

ratios in PBMC medium. The plates were incubated for 48 h. Endpoint was assayed as previously explained.

For the IFN-γ blocking experiments, tumor cells were seeded as explained before, and in addition to the PBMCs and HER2-TCB/CAR Ts, either 20 μg/ml of anti-human IFN-γ (#506512, Biolegend) or mouse IgG control (#400123, Biolegend) was added in the coculture. Experiment lasted 72 or 48 h respectively and endpoint was assayed as previously explained.

**3D organoid assay**. Tumor cells were seeded in 48-well plates (10³ cells/well) in a drop of 20 μl of matrigel (#356235, Corning). Each drop was dispensed in the center of the well and incubated for 15 min at RT. After matrigel was solidified, 250 μl of 3D breast tumor organoid media[35], were added to each well. Media was replaced twice a week and 3D formation was assessed after 15 days. Organoids per well were counted and assumed that each of them consisted of approximately 50 cells. Organoid media was removed and 3D structures were cocultured with PBMCs at a 2:1 ratio in PBMC media and treated with HER2-TCB at 1 nM for 72 h.

For the IFN-γ blockade experiment, same assay was followed but adding either 20 μg/ml of anti-human IFN-γ or mouse IgG control to the coculture with PBMCs.

At the endpoint, organoids were disaggregated by adding 500 μl of trypsin for 30 min at 37 °C. Then, cells were collected and incubated for 30 min on ice to liquify matrigel. Fully disaggregated organoids were washed and stained as previously explained. Number of alive cells was analyzed with FlowJo software by means of EpCAM counts. Gating strategy for tumor EpCAM⁺ counts is shown in Supplementary Fig. 8.

**Flow cytometry**. Cells were harvested with StemPro Accutase (#A1110501, Gibco) and resuspended in FACS buffer. Twenty minute later, samples were centrifuged and cells were incubated for 30 min with the specified antibody. After a wash and Zombie Aqua staining, samples were acquired on LSR Fortessa. The following antibodies were used: hPDL1 (#329736), hCD80 (#305221), hCD86 (#305425), hGalectin-9 (#348905), hB7-H3 (#351003), hB7-H4 (#358103), hHVEM (#318805), hICOS-L (#309403), h41BB-L (#311503), hOX40-L (#316307), hIFNGR1 (#308606), hIFNGR2 (#308504), all from Biolegend at 1:100 dilution. As an isotype control, PE mouse IgG Isotype Ctrl (#400114, Biolegend) was used at 1:100.

In the case of HER2 staining, cells were incubated in FACS buffer for 30 min with Trastuzumab (#180288-69-1, Herceptin, Roche) at 2.5 μg/ml. After two washes with PBS, a secondary conjugated antibody Anti-human Alexa-488 (#A-11013, Invitrogen) was incubated with the cells at a concentration of 1:500 for 30 min. Cells were then washed with 1× PBS and resuspended in Zombie Aqua viability marker and acquired on LSR Fortessa. HER2-TCB binding assay was performed by incubation of cells with 10 nM HER2-TCB for 30 min, followed by the procedure described above.

PD-L2 staining consisted of a primary and secondary antibodies incubation. The incubation with the primary antibody, anti-PD-L2 (#130098525, Miltenyi Biotec), lasted 30 min at 1:11 concentration, and after a wash with 1× PBS, samples were incubated with the secondary antibody anti-biotin (#130113292, Miltenyi Biotec) at 1:50 for 30 min. After a wash and Zombie Aqua staining, samples were acquired on LSR Fortessa using BD FACSDiva software.

The activation marker CD69 (#310914, Biolegend) in CD8⁺ cells (#344712, Biolegend) was used in order to assess T-cell activation after 72 h of coculture with tumoral cells and 67.5 pM HER2-TCB (both at 1:300 concentration).

Ki-67 intracellular staining was performed in PBMCs after 72 h of coculture with BT474 cells using different concentrations of HER2-TCB. First, cells were incubated with zombie aqua for 25 min at 1:500 concentration at room temperature (RT). Then, cells were washed and stained with a CD8 antibody as previously explained. After a wash with 1× PBS, cells were fixed with 200 μl of fresh prepared (1 part of concentrate with 3 parts of diluent) fixation/permeabilization working solution (#00-5523-00, Invitrogen) for 25 min on ice in dark. Supernatant was removed after a centrifugation and cells were permeabilized with 1× permeabilization buffer for 25 min on ice in dark. After supernatant was removed, cells were stained with anti-Ki-67 antibody (#563756, BD Biosciences) in 1× permeabilization buffer at 1:300 concentration for 1 h at RT, washed with 1× PBS and acquired on LSR Fortessa. Flow cytometry data was analyzed with FlowJo software (BD Life Sciences). Gating strategy for T-cell proliferation, activation marker analysis, and for median fluorescence intensity (MFI) measures is shown in Supplementary Fig. 8.

**IFN-γ cytotoxicity assays**. Tumor cells were seeded in flat bottom 96-well plates (0.01 × 10⁶ cells/well). After 24 h cells were treated with different concentrations of IFN-γ. Treatment lasted for 5 days and cell death was assayed by crystal violet staining of alive cells. Cells were fixed for 30 min with 10% glutaraldehyde, washed, and stained for other 30 min with 0.1% crystal violet (#548-62-9, Sigma-Aldrich). After three washes with water, plates were let dry overnight. For the readout, 100 μl of 10% acetic acid were added to each well and absorbance was read at 560 nm using an Infinite M200 Pro Multimode Microplate Reader (TECAN).

**Antitumor drugs cytotoxic assays**. Tumor cells were seeded as described above. After 24 h cells were treated with different concentrations of paclitaxel (#33069-62-4,

Paclitaxel Hospira, Pfizer), doxorubicin (#23214-92-8, Farmiblastina, Pfizer), or T-DM1 (#1018448-65-1, Kadcyla, Roche). Treatment lasted for 3 days in the case of the chemotherapeutic agents paclitaxel and doxorubicin, and 6 days when treated with T-DM1. Cell death was assayed by crystal violet staining of alive cells and read at the Infinite M200 Pro Multimode Microplate Reader (TECAN).

**Western blot**. For Western blot, protein extracts were isolated by lysing the cells in homemade lysis buffer (130 mM NaCl, 0.01% NP-40, 1% glycerol, 2 mM EDTA pH 8 and 20 mM Tris-HCl pH 7.4), supplemented with phosphatase inhibitors 5 μM β-glycerolphosphate, 5 μM sodium fluoride, 1 μM sodium orthovanadate and cOmplete™, EDTA-free protease inhibitor cocktail (#COEDTAF-RO, Sigma-Aldrich, 1 tablet per 10 ml lysis buffer). Protein extracts were sonicated for 10 s at 4.5 V to break the cell apart. Tubes were centrifuged 19,000 ×g 10 min and supernatant was collected.

Protein lysates were resolved by sodium dodecyl sulfate polyacrylamide gel electrophoresis and then transferred to a 0.45 μm nitrocellulose membrane (#10600002, GE Healthcare Biosciences). Totally, 20–30 μg of protein lysate was loaded per experiment. Membranes were incubated with 5% BSA or 5% non-fat milk in TBS-T (1× tris-buffered saline with 0.1% tween 20 (#P7949, Sigma-Aldrich)). After blocking, membranes were incubated overnight with primary antibodies.

After washing, membranes were incubated with horseradish peroxidase-conjugates antibodies (GE Healthcare) for 1 h. Membranes were developed with Immobilon Western Chemiluminescent HRP Substrate (#WBKLS0500, Millipore) and protein bands were visualized in Amersham™ Imager 600 (GE Life Sciences).

Antibodies used were: HER2 (#AM134, Biogenex), JAK1 (#3344, Cell Signaling Technology (CST)), JAK2 (#3230, CST), pSTAT1 (#9167, CST), STAT1 (#9172, CST), IRF1 (#sc-497, Santa Cruz Biotechnology (SC)), and GAPDH (#ab128915, Abcam). All antibodies were used at 1:1000 concentration in 5% BSA except GAPDH (1:5000). Quantification of protein levels was done with ImageJ (National Institutes of Health). Quantifications are the result of ≥3 independent biological replicates. All original Western blots are provided in the Source data file.

**RNA isolation and qRT-PCR**. Total RNA was isolated from adherent cells by using RNeasy Mini Kit (#74106, Qiagen) according to the manufacturer's protocol. RNA was eluted in RNase-free water and quantified using NanoDrop™ 2000 spectro-photometer (Thermo Fisher Scientific).

cDNA was prepared from 1 μg template RNA using the high capacity cDNA reverse transcription Kit (#4368813, Applied Biosystems) according to the manufacturer's protocol.

Real-time quantification of transcript abundance was determined by qRT-PCR using the TaqMan Gene Expression probes (Applied Biosystems) and TaqMan Universal Master Mix II (#4440039, Applied Biosystems), in 384-well plates in 7900HT Fast Real-Time PCR System (Applied Biosystems), following the manufacturer's protocol.

The following TaqMan probes were used: *TIMP2* (Hs00234278_m1), *IFNGR1* (Hs00988304_m1), *IFNGR2* (Hs00194264_m1), *JAK1* (Hs01026983_m1), *JAK2* (Hs01078136_m1), *STAT1* (Hs01013996_m1), *GAPDH* (Hs02758991_g1). Data was analyzed with sodium dodecyl sulfate (SDS) software, RQ Manager, and DataAssist software (Applied Biosystems), using the $2^{-\Delta CT}$ method. *GAPDH* was used as an endogenous control.

**Drugs targeting epigenetic modifications**. Cells were treated with 5 μM of different drugs for 48 h: 5-Azacytidine (#A2385, Sigma-Aldrich), 3-Deazaneplanocin A (DZNEP, #HY-10442, MedChemExpress) or Trichostatin A (TSA, #HY-15144, MedChemExpress). Then, RNA was isolated and subjected for *JAK2* expression by RT-qPCR.

**Chromatin immunoprecipitation (ChIP)**. Indicated cells were grown to 70% confluence, collected, and subsequently cross-linked with 1% formaldehyde shaking at 37 °C temperature for 10 min. Reaction was quenched by incubating the samples with 125 mM Glycine (#BP381, Fisher Scientific) for 5 min. Cells were pellet at 5 × 10⁶ cells/vial and stored at −80 °C.

Cell pellets were resuspended in SDS lysis buffer (1% SDS, 10 mM EDTA, 50 mM Tris pH 8) with 1:200 Protease Inhibitor Cocktail Set III (#535140, Merck Millipore) for 30 min on ice. Samples were then sonicated to generate fragments of DNA between 100 and 600 bp with the Bioruptor (Diagenode). After 20 min on ice, samples were centrifuged at 19,000 × g and supernatant was diluted 1/10 with Dilution buffer (0.01% SDS, 1.1% Triton X-100, 1.2 mM EDTA, 16.7 mM Tris pH 8, 167 mM NaCl), in order to decrease concentration of SDS. Samples were incubated with 10 μl of Dynabeads protein A (#10002D, Invitrogen) and 1 μg of irrelevant antibody Rabbit IgG (#I8140, Sigma-Aldrich) per IP in that sample, as a preclearing. Incubations lasted for 3 h rotating at 4 °C. Magnets were used to discard the beads, and the samples were separated per IP, saving 10% for the input. Totally, 3 μg of corresponding antibody was added at each tube and samples were incubated overnight rotating end over end at 4 °C. The antibodies used were: Rabbit IgG, anti-H3K27Me3 (#07-449, Merck Millipore) and anti-H3K27Ac (#ab4729, Abcam).

Samples were incubated with 50 µl of prewashed dynabeads and incubated 3 h rotating at 4 °C. Dynabeads were then washed 3 times with low salt and 3 times with high salt buffer (0.1% SDS, 1% Triton X-100, 2 mM EDTA, 20 mM Tris pH 8, and 150 or 500 mM NaCl respectively) and 2 times with LiCl buffer (250 mM LiCl, 1% NP-40, 1% NaDOC, 1 mM EDTA, 10 mM Tris pH 8). Samples were then incubated with 48 µl of elution buffer (0.4% SDS, 5 mM EDTA, 10 mM Tris pH 8, 300 mM NaCl) supplemented with 2 µl proteinase K (#RPTOTKSOL, Roche). Then, samples were incubated shaking 1 h at 55 °C and subsequently overnight at 65 °C. Input samples were treated the same way. DNA was purified from the eluted samples with the MinElute PCR Purification Kit (#28006, Qiagen) following the manufacturer's instructions.

Finally, qPCR was performed with the ChIP samples using SYBR green reagent (#733-1390, Quantabio). A 179-bp segment of the *JAK2* promoter was amplified with the following primers: 5′-GGATGTGAGTGGGAGCTGAG-3′ (sense) and 5′-GAGATAACACCCACCGGCTA-3′ (antisense). Data shown is the result of normalizing the specific signal of each antibody (normalized to the IgG control signal) of BT-R or BT-RG to the parental BT474 cells.

**Humanized xenograft models.** In the PBMCs humanized xenograft models, NSG mice were injected orthotopically in two flanks with $6.5 \times 10^6$ or $10^7$ tumor cells in 100 µl of 1:1 PBS:matrigel. Once tumor size reached a specified volume, animals were intraperitoneally injected with $10^7$ PBMCs obtained from healthy donors. After 24 h, animals started treatment and were treated biweekly with HER2-TCB (0.125 mg/kg) or vehicle intravenously.

In the HER2-CAR in vivo experiment, NSG mice were injected orthotopically with $10^7$ tumor cells. Once tumors were around 200-300 mm³, animals were intraperitoneally treated once with $3 \times 10^6$ HER2-CAR T-positive cells.

To obtain immunodeficient mice with a reconstituted human immune system, CD34+ cells were purified from human cord blood obtained through the Blood and Tissue Bank of Catalonia. Blood was diluted 1:2 with 1× PBS + 2 mM EDTA and transferred to a 50 ml falcon tube with 15 ml of Ficoll–Paque PREMIUM (#70-1440-02, GE Healthcare), following the manufacturer's manual. After obtaining the mononuclear cells, resting RBC were lysed with 1× RBC lysis buffer for 4 min. CD34+ cells were purified by negative selection by incubating the mononuclear cells with EasySep Human Progenitor Cell Enrichment Cocktail with Platelet Depletion (#19356, StemCell Technologies), following manufacturer's protocol. Purity of the remaining cell mix was checked with anti-human CD34 (#60013, StemCell) and anti-human CD45 (#304008, Biolegend) staining at 1:300 concentration in FACS buffer. Samples were acquired in LSR Fortessa and percentage of CD34 and CD45 cells were analyzed in FlowJo. Obtained cells were frozen with Cryostor CS10 at −80 °C.

CD34+ cells were injected intravenously in 5-week-old female NSG mice previously treated with busulfan (15 mg/kg) to remove the hematopoietic system of the mice. After 4–5 months, levels of humanization were checked by flow cytometry and mice with more than 30% of human CD45+ cells in blood were used for in vivo experiments. Totally, $10^7$ tumor cells were orthotopically implanted per flank, and once these reached ~200 mm³, animals were randomized and treated biweekly with HER2-TCB (0.25 mg/kg) or vehicle (intravenously).

At the end of the experiments, tumors were analyzed. Tumors were cut into small pieces and divided into samples for IHC, protein, flow cytometry, or reinjection. Samples for IHC were fixed and embedded in paraffin. Samples for western blot were incubated with lysis buffer, supplemented with phosphatase and protease inhibitors, in BashingBead lysis tubes (#S6003, Zymo Research) and homogenized in Precellys Evolution Homogenizer (Bertin Technologies).

Samples for flow cytometry and reinjection were digested in 300 U/ml collagenase IA (#C2674, Sigmsa-Aldrich) and 100 U/ml hialuronidase IS (#H3506, Sigma-Aldrich) in DMEM F-12 medium. After 1 h of incubation at 37 °C with shaking at 10 × g, the mixture was filtered through 100 µm strainers. RBC were lysed with 1× RBC for 5 min RT. After a wash with 1× PBS, samples were counted and either reinjected or stained for HER2 and EpCAM as previously explained and acquired on LSR Fortessa. Data were analyzed with FlowJo software (BD Life Sciences). Gating strategy for MFI measures is shown in Supplementary Fig. 8.

All mice in this study were kept within Home Office limits of 22 °C ± 2 °C, 55–65% humidity and run on a 12 h light/dark cycle that runs from 8 a.m. to 8 p.m.

**Generation of in vivo resistant cells.** In the generation of BT474 resistant to HER2-TCB, the humanized PBMCs xenograft model was used. NSG mice were injected orthotopically with $10^7$ tumor cells. Once tumor size reached 300 mm³, animals were intraperitoneally injected with $10^7$ PBMCs obtained from healthy donors. After 24 h, animals started the treatment and were treated biweekly with an increasing concentration of HER2-TCB or vehicle intravenously. HER2-TCB treatments started from 0.0325 mg/kg, and concentration was gradually increased until reaching 0.25 mg/kg. After two passages, resistant tumors (termed as BT-vR) were obtained.

In the case of the generation of PDX118 resistant model to HER2-TCB, CD34+ humanized xenograft model was used. Humanized mice containing >30% hCD45+ in peripheral blood were orthotopically implanted with $10^7$ tumor cells. Once tumors reached 300 mm³, animals started biweekly treatment with an increasing concentration of HER2-TCB or vehicle intravenously. HER2-TCB treatments started from 0.0325 mg/kg, and concentration was gradually increased until

reaching 0.25 mg/kg. After two passages, resistant tumors (termed as 118-vR) were obtained.

**Immunohistochemistry.** The following primary monoclonal antibodies were used: anti-IRF1 (#HPA063131, Atlas Antibodies) and anti-CD3 (#790-4341, Ventana Medical Systems (Ventana)). For immunohistochemistry, fixed tissue samples embedded in paraffin were sectioned at 4 µm thickness. Sections were heated at 60 °C, deparaffinized with xylene and hydrated with two steps of incubation with different dilutions of ethanol.

When stained with anti-IRF1, antigen retrieval was performed by boiling the samples for 20 min in citrate buffer pH 6 (#S2369, Agilent). Endogenous peroxidase was blocked by incubating the samples with 3% peroxide hydrogen (#108597, Merck Millipore) diluted in absolute methanol for 20 min. Slides were also blocked with 3% BSA in 1× PBS for 10 min. Samples were then incubated overnight with the primary antibody anti-IRF1 diluted 1:650 in EnVision FLEX Antibody Diluent (#K8006, Agilent). Next, the slides were incubated with EnVision System-HRP labeled polymer anti-rabbit secondary antibody (#K4003, Agilent). Samples were then stained with DAB substrate chromogen (#K3468, Agilent) for 1–4 min and counterstained with harris hematoxylin (#H3404, Vector Laboratories) for 2 min, followed by dehydration with ethanol and xylene, and finally mounted in DPX.

Immunohistochemical staining of CD3 was performed using a Discovery ULTRA autostainer (Ventana). Heat-induced antigen retrieval was executed using Cell Conditioning 1 (#950-124 Ventana) for 40 min at 95 °C. Endogenous peroxidase block was performed with the CM inhibitor from the ChromoMap DAB kit (#760-159, Ventana) for 8 min. Then, the anti-CD3 primary antibody, ready to use, was applied 32 min at 36 °C. Next, samples were incubated for 8 min with detection kit UltraMap anti-Rabbit HRP (#760-4315, Ventana). Reactions were detected using the ChromoMap DAB Kit. Finally, the slides were counterstained with Haematoxylin II (#790-2208, Ventana) 8 min and Bluing Reagent (#760-2037, Ventana) 4 min, followed by dehydration with ethanol and xylene, and mounted in DPX.

Slides were scanned in the NanoZoomer 2.0-HT slide scanner (Hamamatsu Photonics) and visualized in the NDP.view2 software (Hamamatsu Photonics).

**Cytokine array.** Detection of 80 cytokines (ENA-78, G-CSF, GM-CSF, Gro a/b/g, CXCL1, CCL1, IL-1 alpha, IL-1 beta, IL-2, IL-3, IL-4, IL-5, IL-6, IL-7, IL-8, IL-10, IL-12, IL-13, IL-15, IFN-gamma, MCP-1, MCP-2, MCP-3, M-CSF, CCL22, MIG, MIP-1 beta, MIP-1 delta, RANTES, SCF, SDF-1 alpha, CCL17, TGF beta, TNF alpha, TNF beta, EGF, IGF-1, Angiogenin, OSM, TPO, VEGF-A, PDGF-BB, Leptin, BDNF, CXCL13, CCL23, CCL11, CCL24, CCL26, FGF-4, FGF-6, FGF-7, FGF-9, FLT-3 ligand, Fractalkine, GCP-2, GDNF, HGF, IGFBP-1, IGFBP-2, IGFBP-3, IGFBP-4, IL-16, CXCL10, LIF, LIGHT, MCP-4, MIF, MIP-3 alpha, NAP-2, NT-3, NT-4, OPN, OPG, PARC, PLGF, TGF beta 2, TGF beta 3, TIMP-1, and TIMP-2) was conducted in culture supernatants of untreated and treated BT474 and BT-R cells. Cells were seeded at $1 \times 10^6$ cells per 6 mL of PBMC medium alone or in coculture with a ratio 3:1 of PBMCs and 67.5 pM of HER2-TCB. After 48 h of coculture, supernatant was harvested and frozen. In order to detect differences in parental and resistant cells cytokine secretomes, a human cytokine array (#AAH-CYT-5, RayBiotech) was used with these supernatants following the manufacturer's recommendations. Quantification of cytokine levels was done with ImageJ following manufacturer's instructions.

**ELISA.** In order to assay TIMP2 release, a TIMP2 ELISA (#DY971, R&DSystems) was used with the same supernatants as specified above, following the manufacturer's manual.

In the case of IFN-γ ELISA, supernatants from treated BT474 and BT-R were obtained at the conditions described above, with different timepoints (24, 48, and 72 h). The culture supernatants were used to perform a human interferon-gamma ELISA (#31673539, Immunotools), following manufacturer's instructions.

**Granzyme B activity.** Target cells (BT474 or BT-R) were seeded, $0.25 \times 10^6$ in 60 mm plates (#430166, Corning), cocultured with PBMCs at a PBMC:Target cell ratio of 3:1 and treated with HER2-TCB at a concentration of 67.5 pM for 72 h. After 72 h, tumor cells and PBMCs were harvested and lysed with 100 µl of lysis buffer. Lysed cells were centrifuged at 21,000×g for 10 min at 4 °C to pellet cell nuclei and other cell debris. Supernatants were harvested and assayed for protease activity. Reaction was performed in a non-treated 96-well-plate (#442404, Thermo Fisher Scientific). Each well contained 25 µl of lysis supernatant, granzyme B substrate Ac-IEPD-pNA (#368057, Sigma-Aldrich) at a final concentration of 300 µM and reaction buffer (0.1 M HEPES pH 7.0, 0.3 M NaCl, 1 mM EDTA) in a total volume of 250 µl/well. Mixtures were incubated at 37 °C overnight and color reaction generated by the cleavage of the pNA substrate was measured at a wavelength of 405 nm with the Infinite M200 PRO (Tecan) plate reader[36].

**Annexin V assay.** Cells were treated with 1 µg/ml of IFN-γ for 5 days. In order to assay the number of apoptotic cells, cells were harvested, washed, and stained with APC-Annexin V (1:20 from stock, #550475, BD Pharmigen) for 15 min RT. After a

wash with 1× PBS, samples were resuspended in 1:500 Propidium Iodine (PI) (#81845, Sigma-Aldrich) viability marker and acquired on LSR Fortessa. Data was analyzed with FlowJo software (BD Life Sciences). Annexin V+ cells were considered as apoptotic cells. Gating strategy is shown in Supplementary Fig. 8.

**Viral tumor cells infections.** For lentivirus production, HEK293T cells were first incubated for 2 h with 25 μM chloroquine (#C6628, Sigma-Aldrich) to increase transfection rate. Cells were then transfected with 1 μg/ml of pMD2.G (#12259, Addgene) envelope expressing plasmid, 1.2 μg/ml of psPAX2 (#12260, Addgene) lentiviral packaging vector, and 1.2 μg/ml of the specific lentiviral vector, using 10 μg/ml of polyethylenimine (PEI) (#24765, Polysciences) as transfection agent. Twenty-four hour after transfection, growth medium was replaced with complete medium. After 48 h, viral particles-containing supernatant was harvested and filtered with 0.45 μm PVDF filters (#SLHV033RS, Millipore).

For infections, target cells were seeded in 6-well plates ($0.5 × 10^6$ cells/well). After 24 h, being the confluence around 75%, tumor cells were incubated with the viral supernatants and 8 μg/ml polybrene (#H9268, Sigma-Aldrich), and centrifuged 45 min at $1000 × g$. After 24 h, medium was replaced with complete medium. Twenty-four hour later, infected cells were selected with either 100 μg/ml hygromicin in BT474 cells, 20 μg/ml blasticidin (#ant-bl, Invivogen) in the case of Lenti-Cas9-2A-Blast (#73310, Addgene) BT474 infected cells, or 1 μg/ml puromycin (#P8833, Sigma-Aldrich) in the rest of infections. Selection was subsequently maintained for one week.

In order to generate resistance to HER2-TCB, BT474 were infected with pBABE-hygro (#1765, Addgene), conferring them resistance to hygromicin.

For silencing, the plasmids were obtained from the lentiviral MISSION shRNA Library: TIMP2 (Clones TRCN0000052433, TRCN0000052434), IFNGR1 (TRCN0000300831, TRCN0000058792, TRCN0000304197), JAK2 (TRCN0000003180), JAK1 (TRCN0000121215, TRCN0000121275), and STAT1 silencing (TRCN0000280021, TRCN0000280024), all from Sigma-Aldrich. As a control, tumor cells were infected with scramble shRNA (#1864, Addgene). To overexpress JAK2 in BT-R cells, JAK2 (V617F)-pcw107v5 was used (#64610, Addgene), and empty vector pcw107 (#62511, Addgene) was used as control.

To generate the BT474 KO IFNGR1 cell line, cells were infected with Lenti-Cas9-2A-Blast (#73310, Addgene). After selected with blasticidin, cells were infected with either a CRISPR gRNA targeting IFNGR1 (#HS5000021477, Sigma) or the LV04 control universal gRNA vector (#CRISPR18, Sigma). These gRNAs confer puromycin resistance and BFP expression, and cells were selected with 1 μg/ml puromycin. To obtain KO IFNGR1 cells, these were stained with hIFNGR1 as explained before, and BFP^high/IFNGR1 negative expressing cells were sorted in FACSAria I Digital Cell Sorter (BD Biosciences), obtaining a pool of cells. Validation of KO IFNGR1 cells was done by IFNGR1 staining.

**RNASeq.** Total RNA was isolated from adherent cells as explained before. Extractions of three independent biological replicas were performed. To determine the total RNA quality and quantity was used Qubit® RNA HS Assay (Life Technologies) and RNA 6000 Nano Assay on a Bioanalyzer 2100 (Agilent). The RNASeq libraries were prepared following the TruSeq®Stranded mRNA LT Sample Prep Kit protocol (Illumina, October 2017). Briefly, total RNA (500 ng) was enriched for the polyA mRNA fraction and fragmented by divalent metal cations at high temperature. In order to achieve the directionality, the second strand cDNA synthesis was performed in the presence of dUTP. The blunt-ended double stranded cDNA was 3′ adenylated and Illumina platform compatible adaptors with unique dual indexes and unique molecular identifiers (Integrated DNA Technologies) were ligated. The ligation product was enriched with 15 PCR cycles and the final library was validated on an Agilent 2100 Bioanalyzer with the DNA 7500 assay (Agilent).

The RNASeq libraries were sequenced on HiSeq 4000 (Illumina) with a read length of 2× 76 bp using HiSeq 4000 SBS kit in a fraction of a HiSeq 4000 PE Cluster kit sequencing flow cell lane. Image analysis, base calling and quality scoring of the run were processed using the manufacturer's software Real Time Analysis (RTA 2.7.7).

RNA-seq reads were mapped against the human reference genome (GRCh38) using STAR/2.5.3a with ENCODE parameters for long RNA. Genes were quantified with RSEM/1.3.0 using the gencode.v29 human annotation. Quality control of the mapping and quantification was performed with 'gtfstats' from GEMtools 1.7.0 (https://gemtools.github.io/). Differential expression analysis was performed with DESeq2/1.18[37] with default parameters. Genes with false-discovery rate < 5% and |shrunken fold change| > 1.5 were considered significant. GSEA was performed with fgsea[38]. PCA was performed with the top 500 most variable genes and the rlog transformed counts. Heatmaps with the top 50 differentially expressed genes (DEGs) were done with the pheatmap R package (https://cran.rproject.org/web/packages/pheatmap/index.html).

Pathway enrichment was assessed through the pre-ranked version of GSEA, and we used gene sets derived from the HALLMARK database[39].

The biological pathways associated with DEGs were explored by using the two GSEA gene sets of Hallmarks. The $p$ values indicate whether DEGs were significantly enriched in a biological pathway compared with the background. Shown are pathways with $p$ values < 0.01.

**HER2-CAR T production.** To produce CAR Ts against HER2, a vector plasmid coding for HER2-CAR was synthesized and cloned into pMSGV-1 retroviral vector (Genscript, Netherlands). Then, stocks of HER2-CAR (pMSGV1-HER2-VL-VH-H8) retrovirus were produced as follows. Firstly, culture plates were coated with Poly-D-Lysine 0.001% w/v in 1× PBS for 1 h at RT, to increase cell attachment. After 1 h, PBS was removed and GP2-293 cells seeded. The day after, cells were transfected with 0.3 μg/ml of envelope plasmid RD-114 (a gift from Alena Gros' Lab, VHIO) and 0.7 μg/ml of transfer plasmid (HER2), with 4.6 μl/ml of Lipofectamine 2000 transfection reagent (#11668, Invitrogen), in DMEM F-12 without supplements. After 8 h, media was changed with complete medium. Three days later, cell supernatant containing retrovirus particles was collected and filtered with 0.45 μm polyvinylidene fluoride (PVDF) filters.

PBMCs were stimulated with 50 ng/ml of α-CD3 (OKT3) (#16-0037-85, Thermo-Fisher) and 300 IU/ml IL-2 (#703892-4, Novartis) in PBMC media for 48 h before transduction. The day before transduction, 6-well plates were coated with 2 ml of 10 μg/ml retronectin (#T100A, Takara) in 1× PBS overnight at 4 °C. The day of transduction, cell supernatant containing retroviral particles was centrifuged in the retronectin-precoated 6-well plates for 2 h at $2000 × g$ at 32 °C. Next, viral supernatant was removed and stimulated PBMCs were added on top in PBMC media with 300 IU/ml IL-2 at a concentration of $2 × 10^6$ PBMCs in 4 ml/transduction well. Plates were centrifuged for 10 min at $500 × g$ at 32 °C. After 48 h, CAR Ts were transferred to cell culture flasks (#156367 and #156499, ThermoFisher Scientific) to continue its expansion. Five days after transduction, CAR expression levels were checked. A minimum of 20% CAR positive cells was used for experiments. After 13 days of expansion, CAR Ts were frozen down in Cryostor CS10 at −80 °C and later used for coculture and in vivo experiments.

**Statistics.** For animal experiments, two-way analysis of variance (ANOVA) with Bonferroni correction posttest was used using Graphpad. In the rest of the cases, unpaired parametric $t$ test was used using Excel. Data were considered significative when $p < 0.05$.

**Reporting summary.** Further information on research design is available in the Nature Research Reporting Summary linked to this article.

## Data availability

The RNAseq data in this study have been deposited in Sequence Read Archive (SRA) database and are accessible through the SRA Bioproject accession code PRJNA674313 (https://www.ncbi.nlm.nih.gov/bioproject/?term=PRJNA674313). Source data are available as a Source Data file. The remaining data are available within the Article, Supplementary Information or available from the authors upon request.

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

## Acknowledgements

This work was supported by Breast Cancer Research Foundation (BCRF-20-08), Instituto de Salud Carlos III Project Reference number AC15/00062 and the EC under the framework of the ERA-NET TRANSCAN-2 initiative co-financed by FEDER, Instituto de Salud Carlos III (CB16/12/00449 and PI19/01181), and Asociación Española Contra el Cáncer (AECC). E.J.A. was funded by the Spanish Government (Juan de la Cierva Formación FJCI-2017-34900). Acknowledgements to the Cellex Foundation for providing research facilities and equipment. This research has been funded by the Comprehensive Program of Cancer Immunotherapy & Immunology (CAIMI) supported by the BBVA Foundation (grant 89/2017).

## Author contributions

E.J.A. and A.M.S. designed and performed most experiments, interpreted, and analyzed the data, and revised the paper. I.R.R. and M.R. produced HER2-CAR T cells and revised the paper. M.E. and A.L. performed in vivo experiments. C.A.F. and A.G. designed the HER2-CAR T. C.K. provided the HER2-TCB, and corrected the paper. J. Arribas designed the study, interpreted the data, and wrote the paper.

## Competing interests

C.K. declares employment, stock ownership, and patents with Roche. A.G. reports receiving funding from Novartis, VCNBiosciences and Merck KGaA, has received speaker honoraria from Roche, and has consulted for Achilles Therapeutics, Neon Therapeutics, Genentech, PACT Pharma and Oxford Immunotherapy. J.A. has received research funds from Roche, Synthon, Menarini, and Molecular Partners and consultancy honoraria from Menarini. The remaining authors declare no competing interests.
