## [Peer Review File · Nature Communications]

REVIEWER COMMENTS

Reviewer #1 (Remarks to the Author):

The authors addressed most of my concerns. However:

- they stated that the animal experiments were performed once. From our point of view, this is not acceptable to ensure the solidity of the results
- they refused to test their hypothesis using the reference CART, i.e. CART19

Lastly, all the panels included in the response to reviewers should be made available to the reader (as suppl, figure) not only to reviewers.

Reviewer #2 (Remarks to the Author):

The authors have made a credible effort to address the critique from the first round of review. The authors have been able to resolve some of the questions and concerns; however, several concerns - related to the technical execution of experiments, the ability to extrapolate the findings to other antigens and indications, as well as the narrative and style of the manuscript - remain.

Ad major criticism 1:

The authors state that control experiments have been conducted to assess alloreactivity in the *in vitro* experiments.

- What remains unclear to this reviewer is whether 'alloreactivity' has been deduced from the *in vitro* data that are shown throughout the manuscript?
- The authors state that in the process of generating resistant tumor cells a "new batch of PBMCs" was used for each round of culture. Does this mean that PBMCs from the same respective donor or a different donor were used for every round of culture?
- What also remains unclear to this reviewer is whether the 6 month co-culture is indeed necessary for obtaining resistant tumor cell lines, and whether the desired phenotype can also be obtained in a shorter period of time (considering clinical relevance: if a tumor needs 6 months to acquire this type of resistance, then this mechanism is practically irrelevant).
- What analyses have been done to come to the conclusion that the desired 'resistant' phenotype cannot be obtained after a short period of time?

Ad major criticism 3:

The data presented in Fig. R12 show only a very low level of T-cell activation. In a typical experiment with a T-cell engaging antibody or a CAR construct, one would expect that the majority (if not all T-cells) are activated and therefore show expression of CD25 and / or CD69. The data shown in Fig. R12 do not make sense to this reviewer.

At major criticism 7:

The authors response is not adequate and this issue is not resolved.

- A balanced discussion would mention alternative mechanisms of resistance that tumor cells employ to evade the therapeutic pressure from T-cell engaging antibodies and / or CAR T-cells (e.g. Singh et al. Cancer Discovery 2019).
- In the abstract alone, there are several statements that are not justified, e.g. the authors state that "disruption of IFN-gamma signaling confers resistance to killing by fully active, correctly engaged T-lymphocytes"; there are no data in the manuscript showing that the T-lymphocytes are correctly engaged (there are no microscopy data to demonstrate T-cell and tumor cell interaction

and adequate immune synapse formation between the two) and thus the statement is not justified; also, the statement that T-cells are “fully active” is not supported by any data shown in the manuscript.

In addition, the authors state that “JAK2 is the component preferably disrupted in tumor cells” and also this statement is not supported by the data as indeed other mechanisms of resistance to CAR T-cell therapy have not been investigated and compared.

- Throughout the manuscript, the authors refer to their T-cell bispecific antibody construct - TCB - as a bispecific T-cell engager (BiTE), which is incorrect as the ‘classic’ BiTE design is that of a single chain variable fragment coupled by a glycine serine linker to a second single chain variable fragment – which is distinct from the architecture of the TCB.

- Another example (randomly drawn from the discussion), is the authors’ statement that the use of JAK2 inhibitors in patients with myelofibrosis and polycythemia vera may impact on cancer immune editing and favor tumor progression. This statement is somewhat out of perspective as it implies that these patients would concomitantly suffer from a solid tumor and receive either a T-cell engaging antibody therapy or CAR T-cell therapy, which is very unlikely.

Reviewer #1 (Remarks to the Author)

The authors addressed most of my concerns.

We thank the reviewer for her/his thoughtful comments on our paper. Admittedly, we have not addressed the totality of her/his concerns, but we did satisfactorily address 30 out of the 33 points raised. Below, we further comment on the three remaining points.

1. they stated that the animal experiments were performed once. From our point of view, this is not acceptable to ensure the solidity of the results.

As mentioned in our previous response, the ethical committee on animal experimentation of our institution does not recommend repeating experiments to confirm statistically significant and concordant experiments. In fact, our institution and the funding agencies supporting this work endorse standards of animal welfare and the NCR3 guidelines: Replacement, Reduction and Refinement (*Replace: avoid or replace the use of animals. *Reduce: minimize animal suffering and improve welfare. *Refine: minimize animal suffering and improve welfare).

However, we did repeat the experiment to show that BT-R cells, the main experimental model used in this study. As expected, the result was similar to that shown in Fig. 1D (Fig. RR1 for reviewers).

Fig. RR1 for reviewers, Tumor volumes are represented as averages \pm standard deviations (n=5 and n=8 per BT474 and BT-R arms, respectively).

2. they refused to test their hypothesis using the reference CART, i.e. CART19

As stated in our previous response, it would be undoubtedly interesting to test if our findings on HER2-positive tumors of different origins also extend to leukemic cells targeted by CAR T cells against CD19. However, since we have no experience or reagents in this field these experiments would considerably delay publication.

3. Lastly, all the panels included in the response to reviewers should be made available to the reader (as suppl, figure) not only to reviewers.

The manuscript already includes seven Supplementary Figures, with ~10 panels each. They contain all the relevant information. To keep the length of the manuscript within reasonable limits, we have not included in these Supplementary Figures reiterative or confirmatory data.

Reviewer #2 (Remarks to the Author):

The authors have made a credible effort to address the critique from the first round of review. The authors have been able to resolve some of the questions and concerns; however, several concerns - related to the technical execution of experiments, the ability to extrapolate the findings to other antigens and indications, as well as the narrative and style of the manuscript - remain.

We thank the reviewer for her/his thoughtful comments on our paper. Below, we comment on the remaining points.

1. Ad major criticism 1:

The authors state that control experiments have been conducted to assess alloreactivity in the in vitro experiments.

a. What remains unclear to this reviewer is whether ‘alloreactivity’ has been deduced from the in vitro data that are show throughout the manuscript?

We tested the effect of samples from every PBMCs batch on target cells, with and without HER2-TCB. As shown in Fig. RR2 for reviewers, in the absence of HER2-TCB, PBMCs from different donors had different effects on target cells, leading to killing of 5 - 60% of the cells, presumably by - “alloreactions of different intensities which parallel different degrees of HLA mismatch, as well as T cell fitness that may depend on the donor” -. We only used batches of PBMCs leading to the killing of < 30% of target cells, and normalized data to cells treated with PBMCs alone.

Fig. RR2 for reviewers, BT474 cells were co-cultured with PBMCs from different healthy donors and treated with 125 pm of HER2-TCB (PBMC:target cell ratio 1:1) for 72h as indicated. Then, viable cells were quantified by flow cytometry using EpCAM as a marker. Results are expressed as averages of duplicated determinations.

b. The authors state that in the process of generating resistant tumor cells a “new batch of PBMCs” was used for each round of culture. Does this mean that PBMCs from the same respective donor or a different donor were used for every round of culture?

We used batches from different donors for each round. Samples from each batch were analyzed as described in the previous point, and only batches with low (<30%) killing activity in the absence of TCB were used.

c. What also remains unclear to this reviewer is whether the 6 month co-culture is indeed necessary for obtaining resistant tumor cell lines, and whether the desired phenotype can also be obtained in a shorter period of time (considering clinical relevance: if a tumor needs 6 months to acquire this type of resistance, than this mechanism is practically irrelevant).

d. What analyses have been done to come to the conclusion that the desired 'resistant' phenotype cannot be obtained after a short period of time?

We did control the sensitivity of chronically treated cells at different time points. After two months, the IC50s of parental BT474 and BT-R cells were similar. However, after four months some degree of resistance was observed, which increased after five months (Fig. RR3 for reviewers) and reached a maximum at 6 months (Fig. 1C).

Fig. RR3 for reviewers, Co-cultures of PBMCs with parental BT474 or cells treated for 4 or 5 months with HER2-TCB, were incubated with different concentrations of HER2-TCB (PBMC:target cell ratio 1:1) for 72h. Then, viable cells were quantified by flow cytometry using EpCAM as a marker. Results are expressed as averages \pm standard deviations.

Regarding the clinical relevance of the mechanism of resistance, we would like to call the attention of the reviewer to the fact that in Fig. S6 we showed that resistant cells can be selected in vivo after two rounds of treatment. Confirming the relevance of BT-R cells, in the resistant models generated in vivo, we also observed downmodulation of JAK2.

2. Ad major criticism 3:

The data presented in Fig. R12 show only a very low level of T-cell activation. In a typical experiment with a T-cell engaging antibody or a CAR construct, one would expect that the majority (if not all T-cells) are activated and therefore show expression of CD25 and / or CD69. The data shown in Fig. R12 do not make sense to this reviewer.

Depending on the concentration of the TCB, the % of cells positive for the activation markers CD69 or CD25 varies from 20 to 80% (see, for example, Lehmann et al Clin Cancer Res (2016) 22, 4417, Fig. 1E or Rius-Ruiz et al Sci Transl Med (2018) 10, eaat1445, Fig. 2H). At concentrations of TCB below IC50, such as that used in panel C, the percentages of cells positive for these markers are as low as those shown in Fig. R12C. Regarding results with CAR T cells, the efficiency of transduction of the CAR in this particular experiment was \sim 30%, and that explains the lower percentages in panel D of Fig. R12.

As the reviewer points out, higher doses of TCBs result in higher percentages of cells positive for the activation markers.

3. At major criticism 7:

The authors response is not adequate and this issue is not resolved.

a. A balanced discussion would mention alternative mechanisms of resistance that tumor cells employ to evade the therapeutic pressure from T-cell engaging antibodies and / or CAR T-cells (e.fg. singh et al. Cancer Discovery 2019).

As suggested by the reviewer, we have included a brief discussion on the mechanism of resistance described in the article by Singh, N. et al published earlier this year in Cancer Discovery.

b. In the abstract alone, there are several statements that are not justified, e.g. the authors state that “disruption of IFN-gamma signaling confers resistance to killing by fully active, correctly engaged T-lymphocytes”; there are no data in the manuscript showing that the T-lymphocytes are correctly engaged (there are no microscopy data to demonstrate T-cell and tumor cell interaction and adequate immune synapse formation between the two) and thus the statement is not justified; also, the statement that T-cells are “fully active” is not supported by any data shown in the manuscript.

The results in Fig. 1I, and Fig. S1D show that the levels of total HER2, as well as those of HER2 at the cell surface, are unaltered in BT-R cells. Further, the binding of the HER2-TCB to BT-R cells is also unchanged (Fig. 1J). Thus, we think it is safe to conclude that lymphocytes are correctly engaged to target cells via the HER2-TCB. However, since we have not directly visualized the binding of lymphocytes to BT-R cells, we have changed the statement by removing “correctly engaged” from the sentence pointed by the reviewer.

The results shown in Supplementary Fig. 1F and G, along with the results included in Fig. R12, show that, in the presence of HER2-TCB, parental cells induce: i) lymphocyte proliferation, ii) upregulation of the expression of the activation makers CD25 and CD69 and, iii) activation of granzyme B. Concomitantly parental cells are efficiently killed, we therefore concluded that lymphocytes are fully active in our assays. Since BT-R cells induce the same levels of the same surrogate markers of activation, we concluded that BT-R cells also activate lymphocytes. However, to include the conservative perspective of the reviewer, we have softened the statement and now we refer to “active lymphocytes” instead to “fully active” lymphocytes.

c. In addition, the authors state that “JAK2 is the component preferably disrupted in tumor cells” and also this statement is not supported by the data as indeed other mechanisms of resistance to CAR T-cell therapy have not been investigated and compared.

We have changed the sentence mentioned by the reviewer to “is a component repeatedly disrupted in several independently generated resistant models.” Since three independent models derived from BT474 cells (BT-R, BT-vR and BT-RG) and three derived from a PDX (118-vR, 433-RG, 667-RG) share the downmodulation of JAK2, we think that the sentence now accurately describes our results.

d. Throughout the manuscript, the authors refer to their T-cell bispecific antibody construct - TCB - as a bispecific T-cell engager (BiTE), which is incorrect as the 'classic' BiTE design is that of a single chain variable fragment coupled by a glycine serine linker to a second single chain variable fragment – which is distinct from the architecture of the TCB.

The only instance in which we refer to our TCB as a BiTE is in the first sentence of the introduction. We introduced the term BiTE because Reviewer#3 of the first version of the manuscript suggested that it is equivalent to TCB, in the sense that both BiTEs and TCBs combine the specificities of antibodies against different targets to recruit lymphocytes. To more accurately describe the relationship between TCBs and BiTEs, we have introduced the following sentence in the Introduction:

... T cell bispecific antibodies (TCBs) -which are functionally analogous but structurally different to Bispecific T-cell engagers, BiTEs -...

e. Another example (randomly drawn from the discussion), is the authors' statement that the use of JAK2 inhibitors in patients with myelofibrosis and polycythemia vera may impact on cancer immune editing and favor tumor progression. This statement is somewhat out of perspective as it implies that these patients would concomitantly suffer from a solid tumor and receive either a T-cell engaging antibody therapy or CAR T-cell therapy, which is very unlikely.

We are currently analyzing whether IFN-gamma signaling is also required for the elimination of target cells by lymphocytes through the binding of the TCR to tumor antigens exposed by the MHC I. Preliminary results show that indeed impairment of IFN-gamma signaling prevent killing by lymphocytes engaged to target cells via TCR. Keeping in mind that, during the elimination phase, cancer immunoediting plays a crucial role in the clearance of early tumor lesions, it is logical to conclude that inhibitors of JAK2 may compromise the elimination phase and, thus, favor tumor progression.

REVIEWERS' COMMENTS

Reviewer #2 (Remarks to the Author):

No further comments.